# 3D Multiple-point Statistics Simulations of the Roussillon Continental Pliocene Aquifer using DeeSse

Valentin Dall'Alba[1], Philippe Renard[1], Julien Straubhaar[1], Benoit Issautier[2], Cédric Duvail[3], and Yvan Caballero[2]

[1]Center of Hydrogeology and Geothermics (CHYN), University of Neuchâtel. Rue Emile-Argand 11 CH-2000, Neuchâtel, Switzerland
[2]French Geological Survey (BRGM)
[3]Fugro France SAS, Castries, France

**Correspondence:** Valentin Dall'Alba (valentin.dallalba-arnau@unine.ch)

**Abstract.** This study introduces a novel workflow to model the heterogeneity of complex aquifers using the multiple-point statistics algorithm DeeSse. We illustrate the approach by modeling the Continental Pliocene layer of the Roussillon's aquifer in the region of Perpignan (southern France). When few direct observations are available, statistical inference from field data is difficult if not impossible and traditional geostatistical approaches cannot be applied directly. On the opposite, multiple-point statistics simulations can rely on one or several alternative conceptual geological model provided using training images. But since the spatial arrangement of geological structures is often non-stationary and complex there is a need for methods allowing to describe and account for the non-stationarity in a simple but efficient manner. The main aim of this paper is therefore to propose a workflow, based on the direct sampling algorithm DeeSse, for these situations. The conceptual model is provided by the geologist as a two-dimensional non-stationary training image (TI) in map view displaying the possible organization of the geological structures and their spatial evolution. To control the non-stationarity, a 3D trend map is obtained by solving numerically the diffusivity equation as a proxy to describe the spatial evolution of the sedimentary patterns, from the sources of the sediments to the outlet of the system. A 3D continuous rotation map is estimated from inferred paleo-orientations of the fluvial system. Both trend and orientation maps are derived from geological insights gathered from outcrops and general knowledge of processes occurring in these types of sedimentary environments. Finally, the 3D model is obtained by stacking 2D simulations following the paleo-topography of the aquifer. The vertical facies transition between successive 2D simulations is controlled partly by the borehole data used for conditioning and by a sampling strategy. This strategy accounts for vertical probability of transitions, which are derived from the borehole observations, and works by simulating a set of conditional data points from one layer to the next. This process allows us to bypass the creation of a 3D training image which may be cumbersome, while honoring the observed vertical continuity.

## 1 Introduction

It has been shown, for example by Naranjo-Fernández et al. (2018), that accounting for heterogeneity is an important step in producing realistic hydrogeological models and to properly manage the water resource, especially in a context of global

climatic changes. The present study proposes a new multivariate workflow, using a multiple-point statistics (MPS) approach, to model the spatial heterogeneity of complex alluvial aquifers. The workflow is applied to the Roussillon aquifer, which is a

multi-layered system composed of the Marine Pliocene aquifer, the Continental Pliocene aquifer, and the Quaternary aquifer. Located along the southernmost part of the French Mediterranean coast, near the Spanish border, this system is used intensively both for drinkable water and irrigation (Aunay et al., 2006). From its social and economic importance, understanding the aquifer is essential for the authorities to ensure a long term and sustainable management of the resource. Since one of the largest source of uncertainty is the identification of the hydraulic conductivity field, it has been decided to focus on the modeling of

the complex geological heterogeneity of the Continental Pliocene layer. This layer consists of alluvial deposits and presents a high level of internal heterogeneity.

To model the heterogeneity, different geostatistical methods have been developed and used in the last decades (Koltermann and Gorelick, 1996; de Marsily et al., 2005). They were employed in different fields going from risk assessment, resources management, mining or petroleum engineering (Matheron, 1963; Strebelle et al., 2002; de Carvalho et al., 2017). All these

methods aim to model the variables of interest at locations where they have not been measured. Traditional geostatistical methods are based on a covariance or variogram models inferred from the data. Kriging (Matheron, 1963) provides the best linear unbiased estimator, it is fast and produces a smooth interpolation. Multi-Gaussian simulation methods, such as Sequential Gaussian simulation approach (SGS) proposed by Journel (1974), are able to generate random fields, depicting the spatial variability of the variable of interest. Truncated Gaussian simulation methods (TGS or PGS) allow to generate discrete realizations

where the spatial relations between the facies (categories) are derived from one or several underlying multi-Gaussian random fields (Matheron et al., 1987). However, these methods are based on two-point statistics and cannot reproduce some geological features such as the sinuosity of a channel or realistic sedimentological patterns. Hence, they are not always suitable for modeling the expected heterogeneities in geological reservoirs. Multiple-point statistics methods have been developed since the 90's to overcome these limitations. MPS techniques allow to generate random fields reproducing the spatial statistics given in

a training image (TI), which is a conceptual model that integrates the geological knowledge of the area of interest. Moreover, unlike traditional approaches, MPS does not require to define an analytical model to describe the statistical spatial distribution of the variable of interest, instead it infers the model in an implicit way from the TI provided by the user (Hu and Chugunova, 2008).

Many MPS algorithms have been developed over the years. The general principle consists in sequentially populating the

simulation grid while reproducing the patterns (spatial statistics) present in the TI. For example, in SNESIM (Strebelle et al., 2002), the statistics of patterns on a pre-defined geometry are stored in a tree shape database that is built by scanning the whole TI before starting the simulation. Then, the simulation proceeds pixel by pixel, a value is drawn randomly according to probabilities conditioned by the surrounding patterns and computed from the data base. As a consequence, the method is memory consuming and limited to the simulation of categorical variables. In IMPALA (Straubhaar et al., 2011, 2013), the

limitation due to the memory is alleviated by using a list shape database, and non-stationary TIs can also be handled with the use of auxiliary variables (Chugunova and Hu, 2008). In other algorithms, such as FILTERSIM (Zhang et al., 2006), CCSIM (Tahmasebi et al., 2012) or IQSIM (Hoffimann et al., 2017), the simulation grid is filled by directly pasting or quilting patches,

*i.e.* several pixels at a time. FILTERSIM uses a set of filters to reduce the dimension of the problem, whereas CCISM is based on cross-correlation between patches. IQSIM proposes a new approach that bypasses traditional ad-hoc weighting of auxiliary variables. The main drawbacks of patch-based methods is often their difficulty to honour conditioning data.

One of the most flexible MPS algorithms is the Direct Sampling (Mariethoz et al., 2010). It is a pixel-based method, where the simulation of one pixel consists in randomly searching for a pattern in the TI that is similar to the pattern centered on the considered pixel in the simulation grid, and then copying the value of the variable from the TI. It has the advantage of making database creation unnecessary, does not require to compute probability, and can handle patterns of varying geometry. By adapting the way of comparing the patterns in the TI and in the simulation grid, the algorithm is able to deal with categorical and continuous variables, as well as with the joint simulation of multiple variables. In this work, we use the direct sampling algorithm implemented in the DeeSse code (Straubhaar, 2019). It is parallelized and offers many options to constrain the stochastic simulations such as continuous rotation/affinity maps or proportion targets. Finally, by generating an ensemble of realizations, it is then possible to estimate any probability of interest from the different facies maps. More details about the features of the DeeSse code are provided in Meerschman et al. (2013); Straubhaar et al. (2016, 2020).

The choice of a simulation technique to model an aquifer at a regional scale depends from different factors. One important aspect is the amount of data available. When the amount of data is large, it is possible to infer rather accurately the statistics describing the spatial variability from the data. Probability distributions about the different rock types, variograms, and spatial trends can be directly estimated and used in the simulation process. This situation often occurs in the mining industry, for example where very large number of drill holes are made during the exploitation of an ore deposit. The configuration is very different in other situations, such as the Roussillon plain, where only a few boreholes are available for a large study area. It becomes then difficult if not impossible to estimate accurately those statistical parameters from the data set. One has then to rely more heavily on indirect data, geological concepts and analogy with other sites. In these situations, statistical distributions, variograms and orders of magnitude of correlations lengths could be borrow from data bases of similar environments such as those developed by Colombera et al. (2012). The issue with that approach is that the simulations may be constrained only by a few data points and therefore the final variability among the simulations will be excessively large and the geological features will not be properly represented because the field data will not compensate the lack of geological concept in a variogram based geostatistical approach. An object based method would respect better the geological knowledge because the user will have to explicitly define the shape of the objects and this approach could be an interesting solution for these situations with an important data gap. Here, we rather consider the use of MPS. As for the object based approach, it allows integrating directly geological knowledge in the stochastic simulation process.

When using MPS, an important part of the process is the construction of the training image. We first want to note that the conceptual sedimentological models are usually represented in 2D map views or block diagrams and geologists are used to express their understanding of a system by drawing such maps and cross sections. Furthermore remote sensing data or geological maps are widely available and can be used to refine these 2D conceptual models. Accessing 2D training images is therefore easy and simple. However, the standard MPS workflow requires a 3D training image to generate 3D simulations. Getting the 3D training image from 2D concepts is not a simple task. It may require a significant amount of tedious work to

construct manually a 3D training image from the 2D concepts. Therefore, previous research was devoted to the design of MPS algorithms able to use 2D training images directly as input for 3D simulations (Comunian et al., 2012; Cordua et al., 2016).

Here, we propose a simple approach that allows the user to avoid the step of the 3D training image construction. This is not mandatory. If a 3D training image is available, it can easily be used in the workflow, but if it is not available it should not be a limitation as we will illustrate in the paper.

Another very important aspect to take into account at the regional scale are the statistical non-stationarities resulting from geological processes such as the location of the sources of the sediments, their transport, deposition, and so on. The application of MPS to a real case requires more than just an efficient MPS code and a good training image, it also requires to develop a methodology and a workflow to account for all those aspects.

The aim of this paper is therefore to introduce a global workflow allowing to incorporate most of the available geological knowledge into a plausible heterogeneity model and to illustrate the method on the Roussillon plain. This approach is generic and can be applied to any other case where the available data are scarce compared to the geological knowledge. The workflow includes a series of steps that are described in detail in the paper. Based on the borehole and geological knowledge of the site, a plan view non-stationary training image displaying the main sedimentological features is designed. In this paper, we limit ourselves to the construction of a 2D training image since there are many situations in which the cross sectional view at the scale of the aquifer is much less well known than the expected spatial organization of the sedimentary layers on a 2D horizontal plane. The vertical transitions are controlled using probability of transitions derived from the boreholes. To control the lateral transitions and non stationarity, a 3D auxiliary map representing a proxy of the evolution of the system from the sources of the sediments to the output is modelled by solving a diffusivity equation. The boundary conditions imposed to the diffusivity equation allow to account for the paleo-input zones and the lateral geometry of the aquifer. In addition, the proposed workflow accounts for the paleo orientations of the sedimentary system and its related uncertainty as inferred from field observations. This work shows that such an approach can be efficient to simulate realistic alluvial systems matching the conceptual knowledge of the system.

The paper is structured as follows : section 2 introduces the background information regarding the geology and hydrogeology of the Roussillon aquifer, and the DeeSse algorithm, section 3 describes the workflow, and section 4 presents the results. The paper ends with a discussion and conclusion.

## 2   Background information

### 2.1   Geology

Located in southern France, this $800 \text{ km}^2$ sedimentary basin is limited by the foothills of the Pyrenees to the south and west, the Corbières massif to the north and the Mediterranean Sea to the east (Fig. 1). This basin originates from the opening of the Gulf of Lion (Oligocene to Miocene) before being largely eroded by the Messinian Salinity Crisis (MSC) (Clauzon et al., 2015). It is with the drying up of the Mediterranean Sea that the Miocene was exposed and eroded, approximately 6 My ago (Lofi et al., 2005). During the Pliocene, the basin was filled up again, with Gilbert delta reworking the sediments generated by the sub-

aerial Messinian unconformity. The sediments grade to wave-dominated deltas (Sandy Marine Pliocene) to fluvial dominated delta with the continental part corresponding to the Continental Pliocene. On top of the stratigraphic pile, Quaternary sediments associated with rivers and lagoon systems have been deposited.

The Pliocene layer is composed of different sandstone units separated by silt and clay layers of low permeability (Duvail, 2012; Aunay et al., 2006). The main sources of sediments came from the weathering of the massifs surrounding the Roussillon's plain. Its depth increases towards the coastline, where its maximum thickness reaches 300 m (Duvail et al., 2005).

Based on field observations, the Continental Pliocene can be considered as a classical fluvial sedimentary system. Near the relief, the association of high energy systems and the large amount of available sediments created alluvial fan deposits, composed of sandstone conglomerates. These alluvial fans have an extent of 1-3 km radius and can be more than 10 m thick. Fans merge together producing larger bodies of 3 to 6 km wide and over 60 m thick. The alluvial fans rapidly evolve to braided river deposits composed of coarse sands and sandstone conglomerates. These braided structures have generally an extent of 100-150 m width and are 1-5 m thick. It appears that these networks can be laterally and vertically well connected, forming very dense and wide objects near their sources. With the decrease of the sedimentary slope, the structures tend to evolve toward meandering river structures. Their deposits are still relatively coarse, yet much more sorted, and well contained within a single channel, their width reaches up to 300 m and their thickness up to 12 m. The connectivity of the river bed deposits is hard to observe either in the vertical or in the horizontal directions. Three other sedimentary elements are also intrinsically developed within the alluvial plain. The first two are the crevasse splay deposits and the levees, which are both directly related to the river's banks flooding dynamic. The last element is the floodplain characterized by a fine grained (silt to shale) sedimentation corresponding to the decanting process of flooding events. In the following, and because we do not consider the deeper Marine Pliocene formations in this paper, we refer to the Continental Pliocene layer/aquifer as Pliocene layer/aquifer.

## 2.2 Hydrogeology

From a hydrogeological perspective, the study area contains two main aquifers; the Quaternary located in the shallow alluvial deposits along the rivers (Agly, Têt and Tech), and the Continental and Marine Pliocene aquifer located deeper and covering the whole basin (Fig. 1). These aquifers are exploited for agriculture and domestic use.

Due to its large extension, both onshore and offshore, the Pliocene's aquifer represents a large water reservoir. However, due to uncertainties related to its properties and recharge processes, the management of this resource is difficult. In the 1960s, the piezometric level was on average 8 m higher as compared to the 2012 data and even artesian at some locations. In recent years and close to the seashore, this exploitation has lowered the groundwater level below sea level during the summer months, when withdrawals are most intense. This situation raises concerns about seawater intrusion risk on the coastal part of the Pliocene aquifer.

As a consequence of climate change, groundwater reserves and recharge may decrease in the near future. For a scenario where the average annual temperature increases by 1.5 C°associated with a decrease in precipitation rate, rivers flow could drop by 40 % over the next 30 years (Chauveau et al., 2013), which will automatically create new stress on the groundwater resource. Considering that the multi-layer Plio-Quaternary Roussillon's aquifer accounts for almost 80 % of the resources used

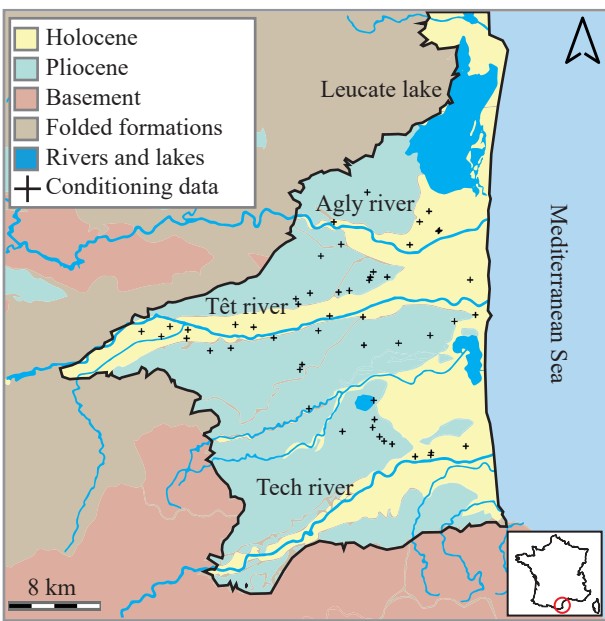

**Figure 1.** Simplified geological map of the Roussillon Plio-Quaternary aquifer.

for drinking waters, there is an urgent need to understand its behavior in order to manage this resource in a sustainable manner
to face global change impacts (Caballero and Ladouche, 2015).

## 2.3  Multiple-point statistics and DeeSse

The essential ingredient of MPS techniques is the TI. The TI is a conceptual model displaying the structures the user wants
to simulate. The use of TI gives flexibility and creativity to the modeller. Unlike some other geostatistical methods such as
two-point statistics, utilization of training image allows specialists from different fields to discuss together about the geometry
and the type of heterogeneity of a model.

A TI can either be stationary or non-stationary. Stationary TIs are easier to use, they display a repetition of patterns with a
homogeneous spatial distribution, *i.e.* the same type of spatial features is present everywhere in the grid. On the opposite, non-
stationary TIs display different kinds of structures depending on the location, they generally include more information and are
more complex. When working with non-stationary TI, some rules must be observed in order to produce realistic simulations.
Since the repetition of patterns is not homogeneous on such TI, one or several auxiliary variables are required to describe
patterns spatial distribution. In the simulation grid, corresponding auxiliary variables are defined to control the spatial location
of the structures that have to be simulated (Chugunova and Hu, 2008). With this information, patterns are not mixed together
when simulated and trend characteristics can be reproduced. Auxiliary variables for the simulation grid are often called trend
maps, because they allow to control the trends of the simulated structures.

A rotation map can also be used to orientate differently the patterns in the simulation grid compared to their orientation in the TI. Hence, the specific spatial features displayed in the TI can follow the same orientation everywhere, which facilitates the construction of the conceptual model, whereas the rotation map defines the local orientation in the simulation grid. Such map consists of angle values defined on the simulation grid for each pixel, the given angle specifying a rotation that must be applied to the TI structures.

As previously mentioned, the DeeSse code is used in this project, which is an implementation of the direct sampling method (Mariethoz et al., 2010). The algorithm is controlled by three main parameters, e.g. $n$- number of neighboring nodes, $f$- scan fraction, $t$- distance threshold. The first one, $n$, defines the maximum number of nodes considered when comparing a pattern in the TI and in the simulation grid. At the beginning of the simulation, these $n$ closest points are likely to be located far away from the simulated point. As the simulation progresses, the density of simulated point increases and the $n$ closest points are starting to be located closer to the central point. This feature enables DeeSse to reproduce structures of all sizes during the simulation, starting with large ones and finishing with small and fine structures (Mariethoz et al., 2010). The second parameter is the threshold value $t$. When comparing patterns during the simulation, DeeSse calculates the pattern similarity between the TI and the simulation grid with a distance value. A perfect match between the patterns represents a distance of zero and completely different patterns correspond to a distance of one. If the distance calculated at the first random position in the TI is larger than the threshold, another point is chosen randomly in the TI and the distance is re-calculated. This is repeated until the value of the distance has reached the threshold or until a perfect match is found, then DeeSse copies the value of the central point found in the TI into the simulation grid. The last parameter $f$, allows to limit the simulation time while conserving realistic patterns reproduction. If a fraction $f$ of the TI is scanned without finding a pattern satisfying the threshold condition $t$, then the best node scanned so far (corresponding to the minimal distance between patterns) is retrieved. The same principles are used for categorical and continuous variables with an adapted definition of the distance. For multivariate simulation, one pattern per variable is considered, with the same central node, and one threshold value per variable.

## 3   Materials and Methods

This section presents the different elements that constitute the proposed MPS workflow. The elements are presented in their chronological order. The section starts by an overview of the workflow before describing the different steps more in detail.

### 3.1   Overview

The first step of the workflow consists in interpreting the geophysical logs and geological field observations to establish the geological concept and build the hard conditioning data set.

The second step consists of converting these observations and concepts into one or several training images. This step is an iterative task, the modeler works with the geologist and they come up with one or several representative TI(s) of the system. In the Roussillon case, the TI used for the Pliocene, is a 2D non-stationary conceptual plan view of an alluvial system composed of

6 sedimentary facies. The TI imposes constraints regarding the geometry of the simulation grid and on the auxiliary information that have to be incorporated in the model.

The third step is the creation of a suitable simulation grid and its associated auxiliary variables. The Roussillon' simulation grid is created based on the bottom topography of the Pliocene, in a flattened space, where 2D simulations can be generated in layers sharing the same age of deposition.

In order to cope with the non-stationary TI, we use two auxiliary variable maps, one for the TI and one for the simulation grid. For the TI, the auxiliary variable is simply the $x$ coordinate re-scaled between zero and one. This variable represents the continuous evolution of the conceptual sedimentary system, advancing from the sediments sources (left side of the TI) to the seashore (right side of the TI). In the simulation grid, we compute the auxiliary variable by solving numerically a diffusivity equation with proper boundary conditions allowing to mimic the general trend of sediments transport from the sediments sources, on the west of the basin, to the coast. The last auxiliary information that is incorporated in the model is rotation maps. The use of the direct sampling algorithm allows us to work with continuous rotation maps, defined for all the nodes of the grid (Mariethoz et al., 2010), whereas classic MPS techniques require to define rotation zones of unique value (de Carvalho et al., 2017). In addition, two continuous rotation maps are used to define the rotation bounds for the simulation with a tolerance of +/- 10° on the rotation values. Theses rotation maps are obtained by kriging data that constrain the paleo-orientations of the main paleo-rivers.

The 3D model is then composed of stacked 2D simulations constrained by 3D auxiliary information. As discussed in the introduction, this approach allows avoiding the construction of a 3D TI that could be cumbersome. To compensate for this choice and to take most of the information available from the hard data set, the 3D grid is created with a rather fine resolution along the $z$ axis (2 m), which corresponds to the smallest body's dimension encountered in the plain. The vertical transition between facies is controlled by simulating additional conditioning data points between two 2D simulations. The values assigned to these sampled points are based on the vertical transition distribution of the facies, inferred from the hard data set. This process allows to bypass the creation of a 3D TI and to simulate with 2D simulation, 3D objects with a realistic $z$ dimension.

The last step consists in generating a set of simulations to characterize the uncertainties. Probability and entropy maps are computed to summarize this information.

## 3.2 Hard data set

Hard data correspond to field observations assigned to cell values in the simulation grid. The hard conditioning data set of the Pliocene model is composed of 52 well logs (lithological, gamma-ray and resistivity logs), which have been described and interpreted in term of sedimentary facies. The boreholes are not homogeneously distributed on the plain but are mainly located along the Têt river and in the central zone of the Roussillon's plain (Fig. 1). Their depths range from 20 m to 150 m and they are on average 77 m deep.

The gamma ray and resistivity logs allowed to identify changes in sedimentary deposits and grain distribution along depth. Sand sediments have a small gamma ray response producing small peaks on the curve, whereas clay sediments produce high response peaks due to their high content in radioactive elements (Serra O. et al., 1975). By analyzing the gamma-ray and

resistivity responses at a certain depth coupling with their vertical evolution, it is possible to identify and assigned a sedimentary facies at a certain depth range. A complete description of the interpretation process of these facies can be found in Duvail (2008).

The hard conditioning data set also incorporates geological information from the geological map of the Roussillon (Genna, 2009). These data correspond to the mapped Pliocene alluvial fan outcrops. We transformed the polygons from the geological map into conditioning data set for the simulation. The facies assigned to these outcrops corresponds to the alluvial fan facies.

The final conditioning set results of 3'500 interpreted points, that are being used during the simulation as hard conditioning data.

## 3.3 Training Images

Based on field observations, well logs analysis and the general understanding of the sedimentary processes composing the Pliocene, three TIs were created (Fig. 2). They correspond to different possible conceptual representations of the Pliocene and were tested using 2D simulations. As discussed by Høyer et al. (2017), the creation of the TI is an iterative task and it is always preferable to compare TIs not only on their structural aspects but also on their associated MPS simulation outputs. This is particularly important when the model includes non stationary TIs and uses complex auxiliary variables, making the simulated patterns difficult to predict from the TIs alone. The three TIs (Fig. 2) describe alluvial systems composed of similar elements and spatial patterns evolution: the system starts from the sediments sources on the upstream side, with alluvial fan deposits, and gradually moves toward the output of the system on the coastal side. In all cases, the facies evolves from braided to meandering river deposits. The three different training images are proposed to test different assumptions concerning the spatial arrangement of the facies at different scales.

The first TI (Fig. 2a) was created based on the visual interpretation of satellite images of the Tagliamento river, which is located in Northern Italy near the town of Udine close to the Slovenia border. The entire channel belt is considered as the main deposition zone. This first TI neither represents the small scale internal structures of the river deposits within the channel belts nor the levee structures. The output of the 2D MPS simulation using this TI results in the creation of too many small braided/meandering river deposits on the plain.

The second TI (Fig. 2b) was created based on a more conventional conceptual representation of braided/meandering river systems. It includes a more complex braided structure and larger meandering river beds. The resulting simulation displays meandering river deposits that are wider than those observed in outcrops in the Roussillon plain. Moreover, the alluvial fan deposits -dark blue facies- are under represented compared to the field observations.

The third TI (Fig. 2c) was obtained by trial and error adjustments. It is composed of six sedimentary facies (Fig. 2d) and displays the evolution of an alluvial system from the mountain (sediments sources) to the seashore, without including the estuary part (Nichols and Fisher, 2007). It is this TI that is used for the next modeling part of the workflow. In this last TI, meandering river channels are represented by a straight shape because our concept here is that the TI does not show the patterns that would be found in a snapshot of a fluvial system in surface but rather represent an integrated view of the sedimentary system through time. This concept is illustrated in Fig. 2e, with the meandering river facies used as an example. At $t = 0$,

the meandering river bed follows one path, controlled by the sedimentary slope and the topography. At $t = 1$ this bed would have laterally migrated, which could cut through the previous one. Finally, at $t = 2$ it is possible to define an area of high river bed occurrences, where all the meandering river bed facies would be located. The last one, $t = 3$, highlights the only possible location where crevasse splay and levee can be preserved, on the borders of the river channel belt.

The final TI is composed of $100 \times 125$ cells with a $100 \times 100$ m dimension. These dimensions are chosen to avoid any affinity (scaling) transformation during the simulation. To represent the possible variability for the position of the transition between braided and meandering river deposits and the distance between the channel belts, the final TI includes some variability on these aspects (Fig. 2c). Note also that the dimension of the crevasse splay deposits increases with the decrease of the sedimentary slope (assuming the sedimentary slope decreases as we move from the sediments sources to the seashore). Finally, the levee facies is incorporated in the TI outside of the meandering river objects. Even if this facies is not recognized in the boreholes data, its spatial location will be constrained by the meandering facies during simulation.

### 3.4 Flattened space simulation grid

With the creation of the TI, the conceptual sedimentation process of the Roussillon's plain is now transferred into a model. The next step consists of creating a suitable simulation grid (SG) for the MPS simulation in accordance with the sedimentation process expressed in the TI. As mentioned before, it is decided to create the 3D model by staking 2D simulations in a transformed grid.

A regular cartesian grid is used for the simulation with the following dimensions: $407 \times 504 \times 125$ cells ($25'641'000$ cells in total) with a cell dimension of $100 \times 100 \times 2$ m. The $z$-axis dimension is defined in order to represent the minimal size of the sedimentary objects that we want to model while the $x$ and $y$ cell dimensions are defined to optimize the resolution of the modelled objects while keeping the computing time reasonable.

Digital elevation maps corresponding to the top and bottom altitudes of the Pliocene (Duvail, 2012) are used to select the active cells of the 3D simulation grid, the final volume of the Pliocene grid is composed of $3'753'230$ active cells (Fig. 3a). Since the TI represents the sedimentary evolution of a fluvial system, the 2D simulations have to be carried out in cells that share the same age of deposition. This requires to transform our 3D grid based on the topography of the bottom layer (Fig. 3b). A vertical shift is applied to each column of cells to flatten the base of the grid to the bottom of the Pliocene. With this transformation (flattened space), we create a 3D grid where it is possible to simulate inside horizontal layer $z_i$ composed of cells sharing approximately the same age of deposition.

### 3.5 Trend Maps

In order to cope with the non-stationary TI, the model has to be constrained with one auxiliary variable map (trend map) for the TI and one for the simulation grid. For the TI, the trend map is simply the $x$ coordinate re-scaled between zero and one and corresponds to the lateral evolution of the fluvial system. This trend map has to be associated with another trend map of similar range for the simulation grid.

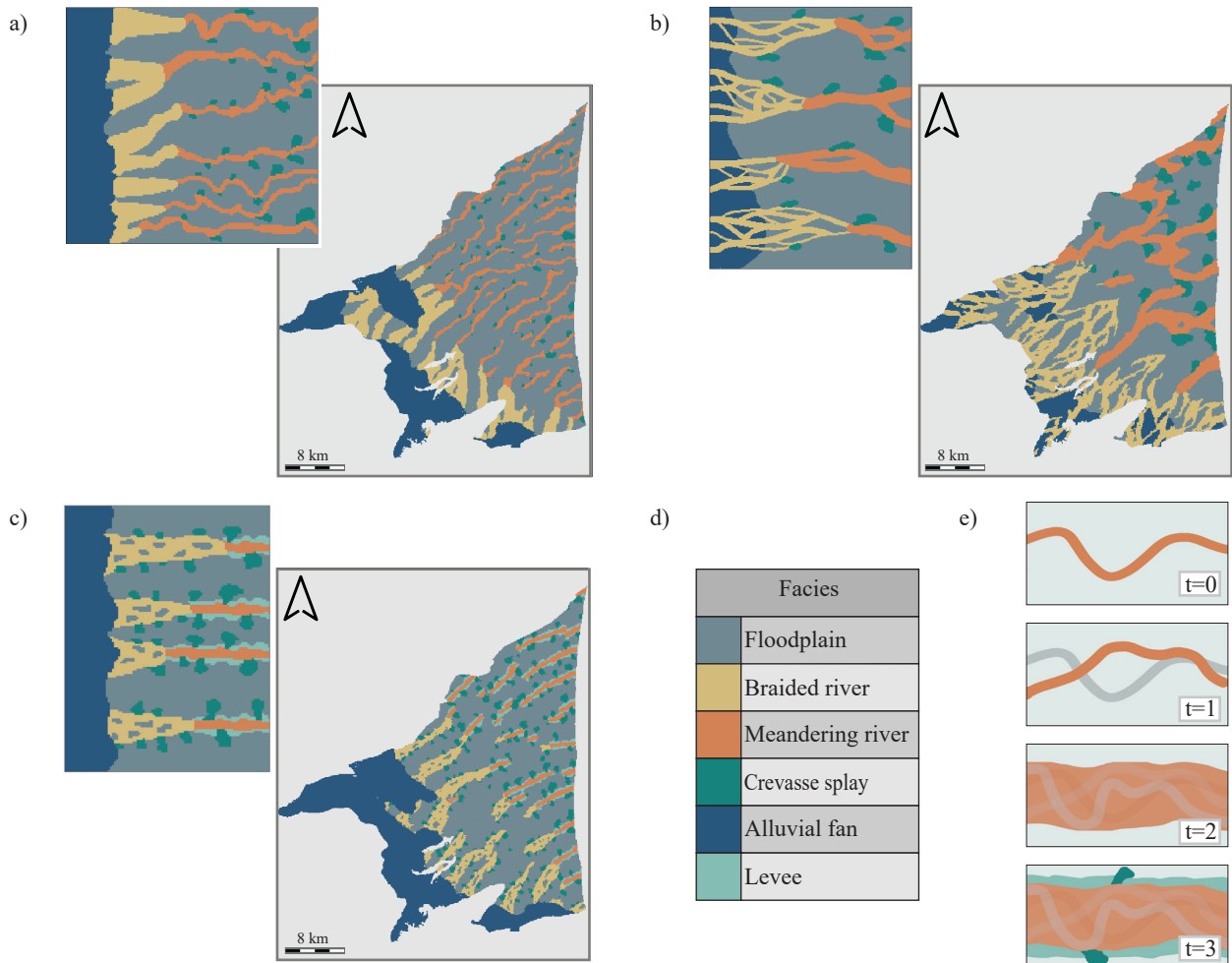

**Figure 2.** Horizontal training images (TIs) associated with their corresponding 2D MPS simulations. a) analog TI derived from satellite images of the Tagliamento river, b) a conceptual analogue TI and c) a TI created based on outcrop descriptions and general knowledge of the Roussillon's plain. d) the six sedimentary facies of the final TI. e) illustration of the sedimentological concept for the creation of the final TI, the aim is to simulate the area of high river bed occurrences (channel belt) rather than individual river channels.

Creating a 3D trend map for the simulation grid is complex due to the geometry of the layers and requires to develop a new approach, different from the one used for the TI. In the flattened space grid, the auxiliary variable, a trend map ranging between $[0-1]$, is computed by solving numerically a diffusivity equation in steady state ($\Delta h = 0$, with $\Delta$ representing the Laplacian operator) for each of the 2D layer composing the 3D grid. The problem is solved using a finite element mesh following the exact geometry of the domain. The boundary conditions are: prescribed values $h(x) = h_0$ corresponding on some parts of the boundary; and $\boldsymbol{\nabla} h(x) \cdot \boldsymbol{n}_x = 0$ on the rest, meaning that the gradient of $h(x)$ should be perpendicular to the vector $\boldsymbol{n}_x$ that is

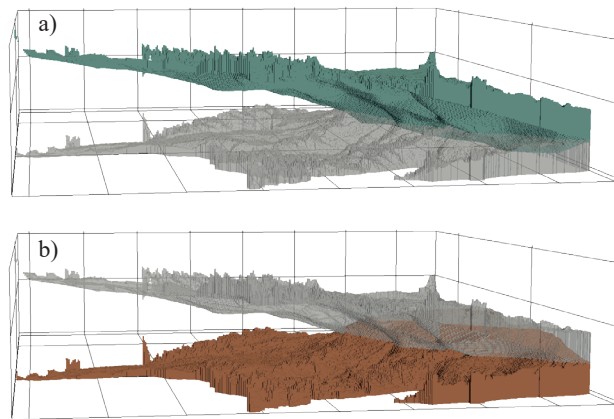

**Figure 3.** a) in dark green the 3D grid of the Pliocene (the grey volume representing the transformed space). b) in dark orange the transformed grid (flattened space) of the Pliocene layer inside which the 2D simulations are simulated (the grey volume representing the original space). The vertical scale is exaggerated in this representation. View from the South of the area toward the North.

normal to the boundary at that location, i.e. the maximum variation of the trend must be parallel to the boundary. This problem is similar to the simulation of the hydraulic heads in a homogeneous confined aquifer for a steady state flow.

After setting the proper boundary conditions – four input zones set to $h_0 = 0$ and one output zone set to $h_1 = 1$ (Fig. 4a) – we obtain the trend map (values ranging between $[0-1]$) by solving numerically the diffusivity equation. The resulting map is a proxy for describing the evolution of the sedimentary system in the SG (Fig. 4b). The four zones with values close to zero correspond to the main paleo-river entrances and to the south relief zone where alluvial fan deposits are known to be present (Fig. 4a). The output zone with values close to one corresponds to the seashore. This method allows to create trend maps

that respect the geometry of the SG and takes into account the paleo river locations. It mimics the general trend of sediment transport from the sediment sources to the coast but it does not constitute an attempt to developing a physically-based model of sedimentation.

This approach is also used to create the vertical sedimentation trend of the plain, corresponding to the progradation of the sedimentary system towards the sea. By simulating 12 representative 2D trend layers (Fig. 4 b) and combining them together vertically (assigning the first trend layer to the layers 0-9, the second trend layer to the layers 10-19 and so on), a complex 3D

trend map is finally obtained. This trend map accounts for both vertical and lateral sedimentation trends that characterize the Roussillon's plain (Fig. 4c).

### 3.6   Rotation Maps

A 2D rotation map is created to control the orientation of the structures in the SG relatively to their orientations in the TI. This

map is built based on data gathered from field observations and interpretations of assumed rivers' paleo-orientations (Fig. 5a).

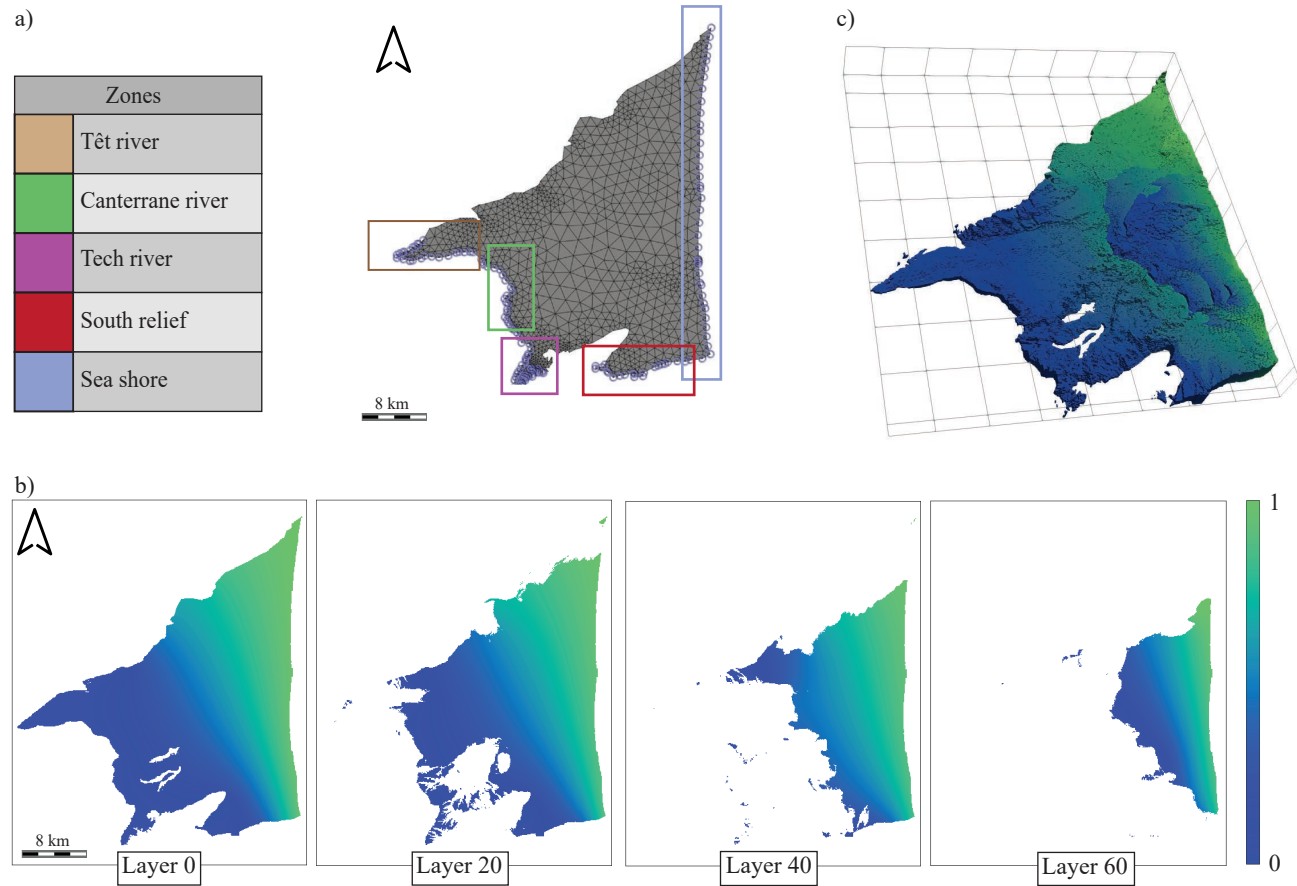

**Figure 4.** a) meshed grid with the four input zones and the one output zone used for the resolution of the diffusivity equation, b) different 2D layers that compose the 3D trend map of the transformed grid, c) top view of the 3D trend map in the transformed space, where the progradation of the trend value towards the seashore is visible.

The main river influx came from the Têt river in the central part of the basin and from the Tech river in the south-west part. Based on these orientations, a fictive rotation point set is created and interpolated using kriging.

The orientation map is based on interpretation and therefore uncertain. DeeSse allows to account for this uncertainty. A tolerance of +/- 10° is considered and added/subtracted to the kriged map to obtain two rotation maps: one with the minimal angle values and one with the maximal angle values (Fig. 5b).

The 3D maps are then created by extruding these rotation values along the $z$-axis, assuming that the variation of the paleo-orientations through time is encompassed within the tolerance values.

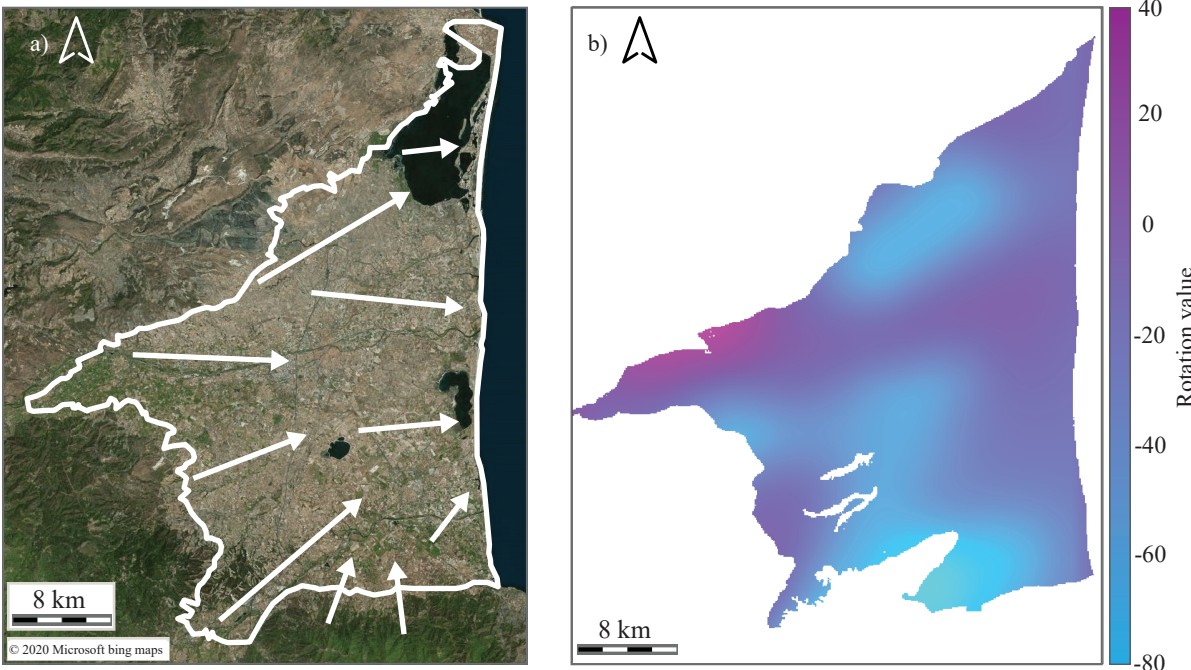

**Figure 5.** a) interpreted orientations of the paleo rivers in the Roussillon's plain. b) interpolated continuous kriged values, where a positive rotation value corresponds to a clock-wise pattern rotation and and a negative rotation value corresponds to an anti-clock-wise pattern rotation.

## 3.7 Vertical transition

To control the vertical transition from one layer to the next one, we developed a simple sampling approach, illustrated in Fig. 6.
The approach starts by simulating the first layer of the transformed grid (layer 0/bottom layer) using only the borehole hard data set as conditioning data. Once this layer is simulated, points are sampled from this layer and propagated as additional (or secondary) hard data for the next layer. The facies value assigned to these points is drawn accordingly to the vertical transition probability between two facies, calculated from the boreholes hard data set.

Three parameters control this method, the first one defines which facies have to be sampled. After some tests, it appears
that concerning the Roussillon case, the best way to control the vertical continuity of the objects of interest is to sample only from three facies: the alluvial fan, the braided river and the meandering river facies. Since the floodplain facies is the most frequent one, sampling this facies at random location leads to an over-representation of the flood plain and tends to bias the MPS simulations. The levee and crevasse splay facies are not sampled in order to avoid to over constrain the structure of the fluvial objects. The second parameter is the sampling rate. It is here fixed at 1 % of the number of simulated cells for each of
the three facies. The last parameter controls the maximum number of successive layers that are simulated using the sampling approach. This mechanism allows to control indirectly the maximal vertical size of the objects. This last parameter is set to

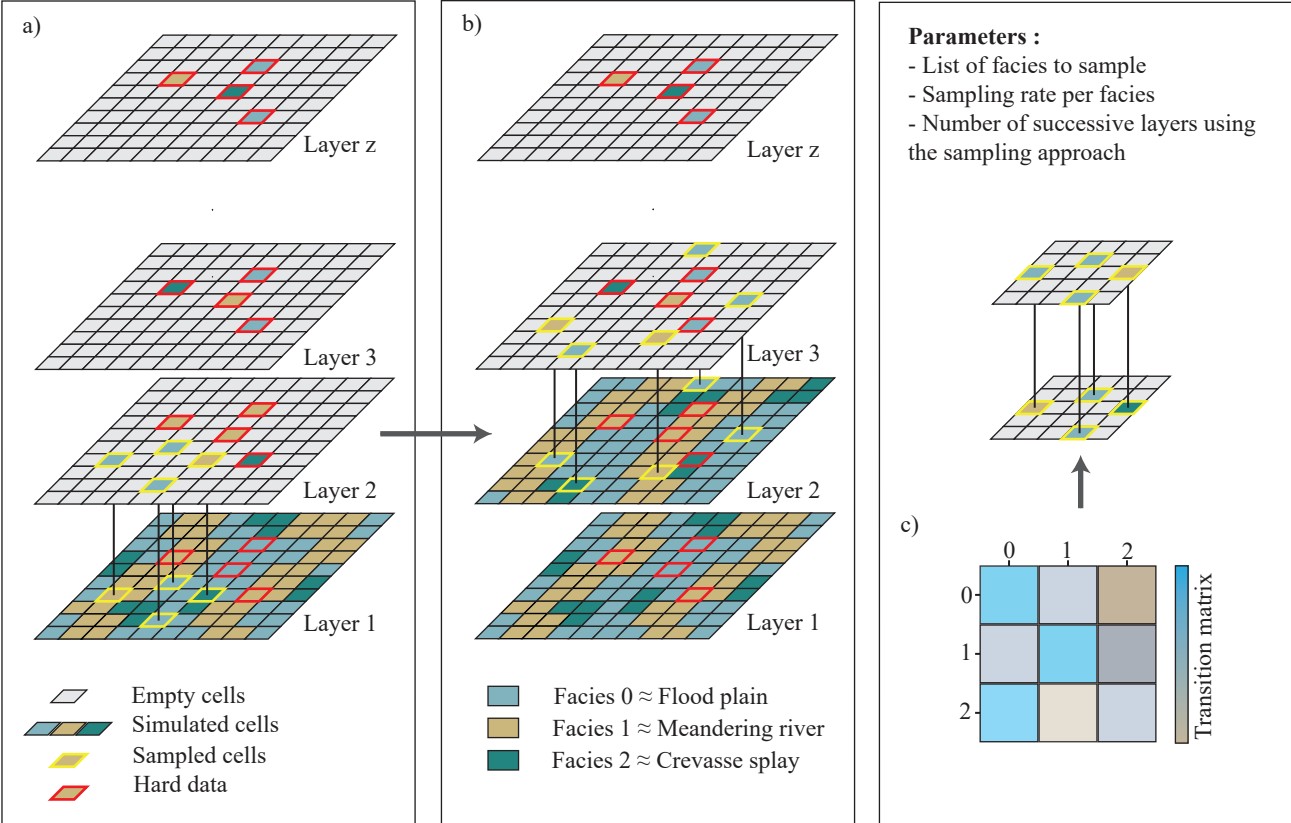

**Figure 6.** a) the first simulation takes place with only the hard data set as conditioning data. Then a set of cells is sampled (in yellow) and used as new conditioning data for the next simulation. b) the process is repeated until the defined number of successive layers is reached. Once reached a simulation takes place without a sampling set. c) the value of the sampled cell is drawn based on the vertical transition matrix, calculated from the borehole data set.

6 for the Roussillon case, meaning that after six successive layers simulated using the approach, the next one will not use secondary sampled hard data.

Without this approach the vertical transition between facies would only be controlled by the hard conditioning data which
are scarce compared to the size of the SG and the number of active cells that compose it.

## 3.8 DeeSse parameters

The main parameters used for the MPS simulation with DeeSse are tested and chosen in order to minimize the simulation time without impairing the quality of the outputs. Two variables are considered: the facies (categorical) and the trend (continuous). The parameters defined for these two properties are the search ellipsoid which allows to limit the size of the pattern, the
360 maximal number of pattern nodes ($n$), the acceptance threshold ($t$) and the scan fraction (f).

The search ellipsoids are identical for both variables and are defined by a radius of 20 cells in $x$ and $y$ axis directions and 0 along $z$ because 2D simulations are performed. The maximal number of nodes is set to 24 for the facies variable and to 5 nodes for the trend variable. The larger number of neighboring nodes for the facies is defined to ensure a proper pattern reproduction during the simulation at both large and fine scales. Since the trend variable is defined for every node of the SG, there is no need to define a large number of nodes for that property. The threshold parameter $t$ that controls the pattern quality reproduction is set to 0.05 for the facies property and to 0.25 for the trend property. Finally, the scanned fraction of the TI is set to 0.75.

Once satisfied with the 3D simulation output, the last step of the approach consists of producing a large number of simulations in order to compare them and to study the uncertainty of the model. The simulations are run on a CPU cluster, allowing to parallelize the computational load between different CPUs.

## 4 Simulation results

The following section presents the results of the workflow and the models obtained with DeeSse. The general aspect of the simulation is first discussed, before focusing on the ensemble statistics results calculated from 50 simulations sets.

Note that MPS validation is still an active research topic. Some tests and approaches are discussed for example by Mariethoz and Caers (2014). However, due to the small number of hard conditioning data, we limit ourselves in this work to analyse the plausibility of the geological patterns in the simulations with respect to the conceptual model, the final geological uncertainty resulting from the model, and some summary statistics.

### 4.1 3D simulation

One simulation is presented in Fig. 7. The first observation is that the model reproduces well the training image patterns over the different layers of the grid. All the main features of the TI are reproduced and the different patterns are globally not mixed between each others as it can be observed in the different horizontal-sections (Fig 7c) . Some discontinuities between the braided and meandering river deposits can be observed. These discontinuities are due to the presence of hard conditioning data, which do not match the pattern locations imposed by the SG trend maps. The continuous rotation maps produce smooth pattern rotations, which could not be obtained with classical zonal rotation. This feature helps to not break the pattern continuity and helps to create realistic simulated shapes.

Regarding the non-stationarity, we can see that the simulated structures successfully follow the trend imposed by the auxiliary variable (Fig. 7c). In particular, the progradation trend imposed to the grid succeeds to reproduce a realistic vertical progradation of the system (Fig. 7b). The alluvial fans, which represents the start of the sedimentary system, gradually move toward the sea as the depth decreases in the model.

One 3D model, composed of $3'753'230$ active cells, is generated in about 15 min on a Intel®Core i7-7700 CPU at 3.6 GHz.

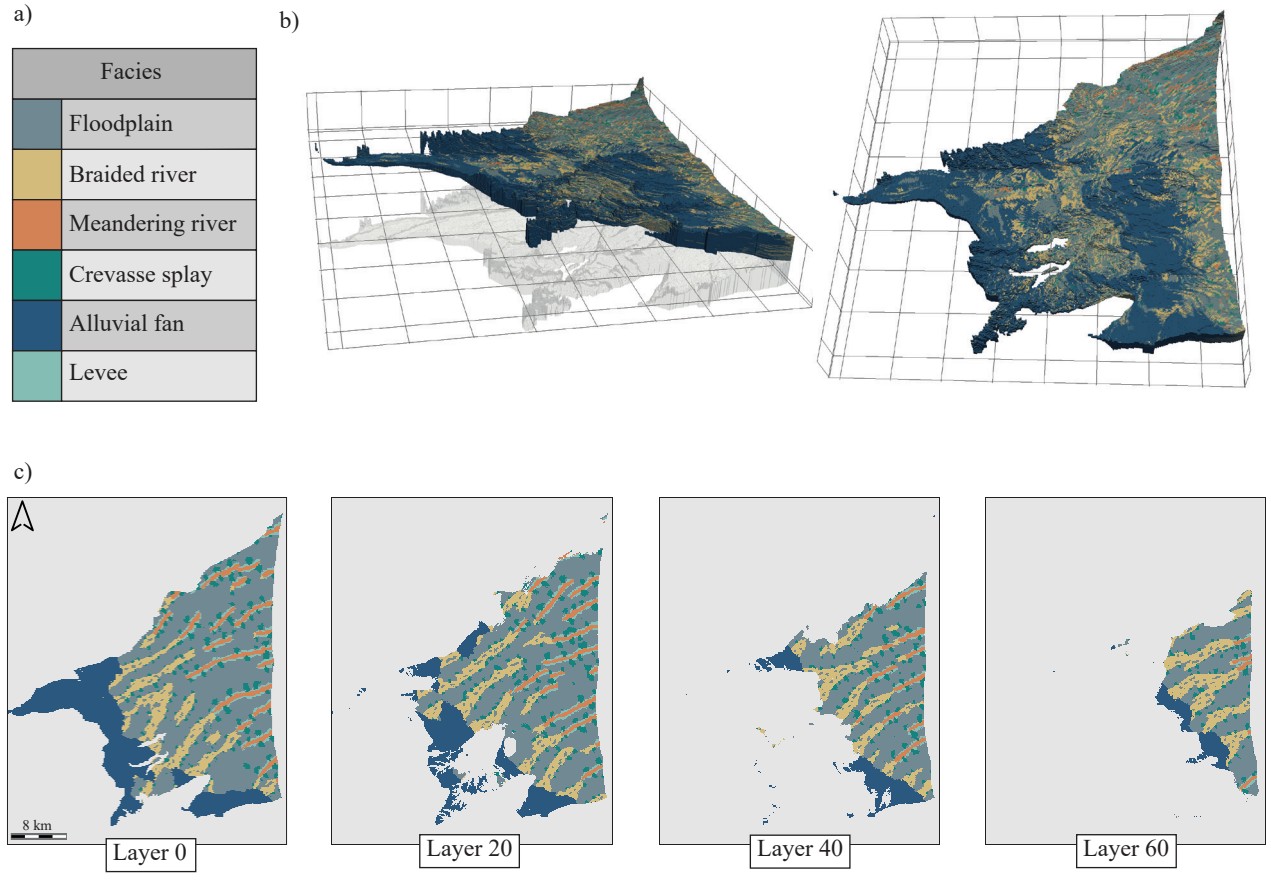

**Figure 7.** a) the six simulated facies, b) one 3D model of the simulations set, in the original 3D grid (left figure), and in the transformed grid (right figure). c) different $z$-layers (horizontal sections in the 3D transformed grid).

## 4.2 Probability Maps

Simulating a large number of realizations enables us to calculate probability maps (Fig. 8) and the pixel wise entropy of the simulations set (Fig. 9).

The probability maps display the probability of facies occurrence at each grid location based on 50 simulations (Fig. 8). If a facies is largely constrained at a spatial location, it is likely that all the simulations will simulate this facies at the same location and thus the probability map will display either very high or very low value at this spatial location. On the contrary, if a facies is less constrained, its probability map will display larger zones of occurrence through the simulations with more moderate values. The zones of extreme values are generally around hard conditioning data locations, which induce zones of low variability near them. These maps help the modeller to understand the model and the associated uncertainties. In this case we are focusing on the uncertainty regarding the shape of the channels and the uncertainty links to their spatial location.

In Fig. 8, we can see that every facies has a different variability behaviour through the ensemble of simulation. However, the facies variability is not influenced by the depth of the simulated layer, every facies displaying the same probability range at different depths.

The alluvial fan facies is the more constrained of all of the simulated facies. It is mainly due to its low spatial variability imposed by the trend maps. The second most constrained facies is the floodplain. It is mainly constrained by the other five facies, which explains its low variability in simulation. These high values are also linked to the dominance in proportion of this facies. For the braided river facies, the spatial probability of occurrence evolves through the layer and is mainly constrained by the hard data, creating zones of high probability. Moreover, the location of these facies are mainly controlled by the trend value of the simulation grid. The same observation can be transposed for the meandering river facies. Finally, the crevasse splay deposits are homogeneously distributed through the layers and the levee facies is only simulated near the meandering river facies.

These maps also highlight the fact that the model is not over-constrained by neither the TI nor the hard data. Focusing on the meandering and braided river facies. We can observe that in some locations the probability maps show high probability values due to the presence of conditioning data. However, this probability decreases proportionally to the distance to the hard data, which demonstrates that the model is not over-constrained by the conditioning data. Moreover, even when a river bed location is constrained with a hard data, the associated spacing with other river bed is not fixed and can fluctuate through the simulation set, which indicates that the TI does not lock the locations of the pattern through the ensemble of simulations. Finally, this analyse of the simulations outputs is satisfying since it shows that the model respect the depositionnal concept expressed by the TI and the trend map.

## 4.3 Entropy Map

The six probability maps are used to calculate the information entropy. The Shannon Entropy was introduced in the theory of information developed by Shannon in the middle of the 20th century (Shannon, 1948) and represents the amount of information carried within a probabilistic distribution. As proposed by Wellmann and Regenauer-lieb (2012), information entropy is an effective tool to visualize uncertainties in a spatial context. The main advantage of the entropy is that it summaries the overall uncertainty contained in a probability distribution with a single number. The entropy is defined as:

$$H = -\sum_{i=1}^{n} p_i \log_n(p_i) \tag{1}$$

where $n$ is the base of the logarithm corresponding to the number of categories (or the number of facies in our case, six) and $p_i$ the probability of occurrence of the $i$-th category. The entropy is maximal and equal to one when all the outcomes have maximum uncertainty and is equal to zero when there is a perfect certainty on the outcome.

The entropy map (Fig. 9) shows that there is little geological uncertainty in the upstream part of the plain where the alluvial fan dominates. Similarly, the uncertainty is rather small in the transition zone between braided and meandering river. The entropy map also reveals that the meandering river facies is mostly constrained around the hard data.

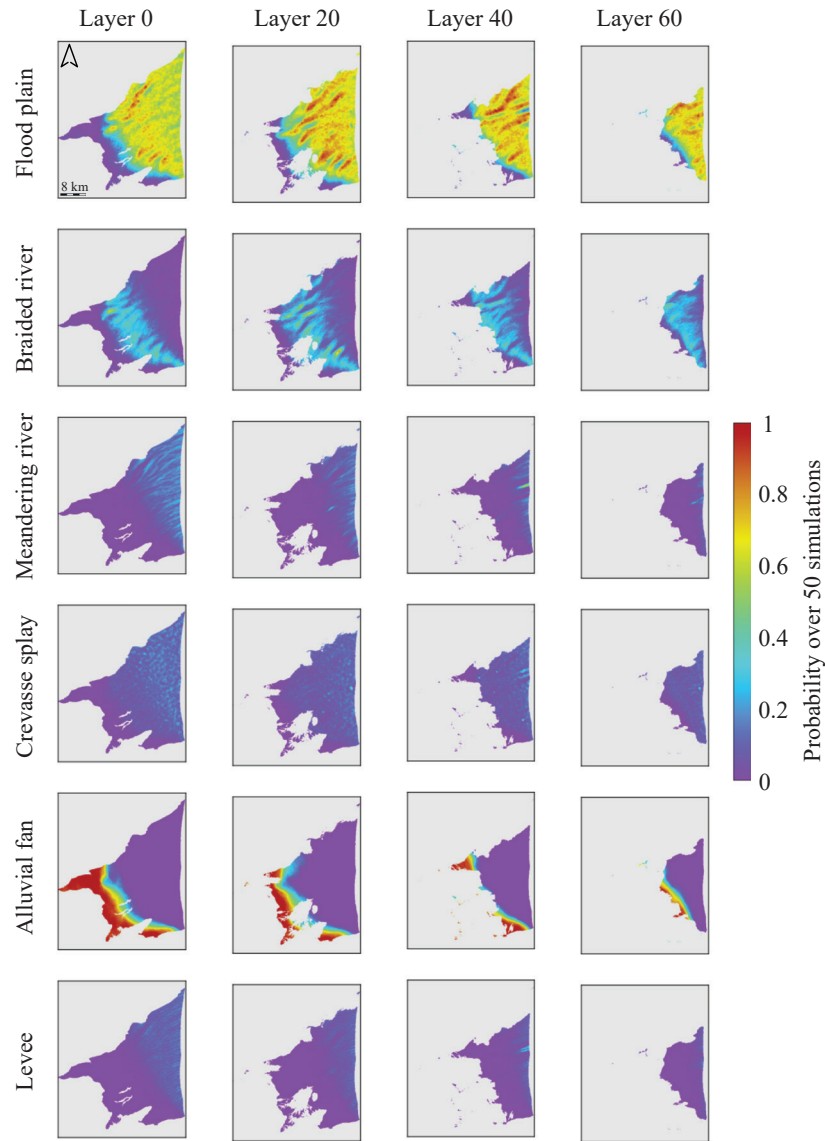

**Figure 8.** Probability maps of the six simulated facies at different depth. The probability maps are calculated over a 50 simulations set.

## 4.4 Facies proportion and vertical transition

Figure 10a compares the proportion of facies from the TI, the hard conditioning data set and two simulation sets, each composed of 50 simulations, the first one with the vertical sampling approach and the second one without it. Overall, the proportion of simulated facies is satisfyingly reproduced when we compare the proportion distributions of the simulation sets against the proportion distribution of the hard data. It appears that the facies proportions are controlled by both the TI and hard data, with

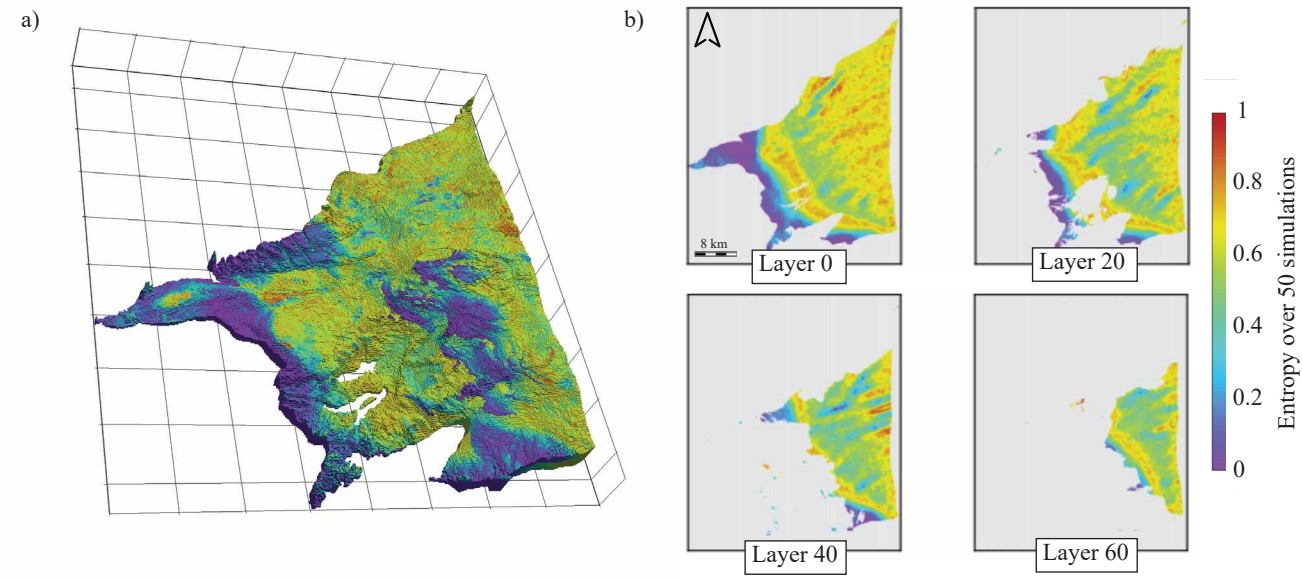

**Figure 9.** Shannon entropy of the model, calculated from the six facies probability maps. a) 3D views in the transformed grid, b) different $z$-layers (horizontal sections in the 3D transformed grid).

the hard data set having a slightly larger influence. This is reflected with the alluvial fan facies, which is less represented in the hard data set - mostly due to the central location on the plain of the majority of the boreholes - and less represented in the model compared to the TI proportion. These facies proportion distributions show the importance of the hard data on the simulation output and the consequence that can arise from a biased hard data set.

To quantify the impact of the vertical sampling strategy, following previous authors, we compared the distributions of the vertical runs (Mood, 1940; Boisvert et al., 2007). To compute this indicator, the 3D grid is decomposed as a set of vertical columns of voxels. A vertical run is then defined as the length of a succession of the same facies values preceded and succeeded by a different facies. By computing the run length on all the columns for a given facies, one can compute the empirical distribution of runs for this facies. The same operation is conducted for all of the facies. In addition, these empirical distributions are also computed on the borehole data. We then compute dissimilarity indices between the simulated and observed distributions for all the facies using a normalized euclidean distance. The closest to zero the dissimilarity value is, the more identical the distributions are and reciprocally (Fig. 10b). The alluvial fan facies is here not represented, because it is under-represented in the hard data set and a reference distribution cannot be inferred from it.

The impact of the sampling approach can also be observed when studying vertical cross-sections along the $x$ and $y$ axis in the transformed grid space (Fig. 11). In Fig. 11a the channels created by the stacking of the braided/meandering facies are vertically disconnected from each other. The sampling approach leads to the creation of vertically connected objects as can be observed in Fig. 11b. With this approach, "channels like" cross-sections can be observed in the simulation results while this was not the case before.

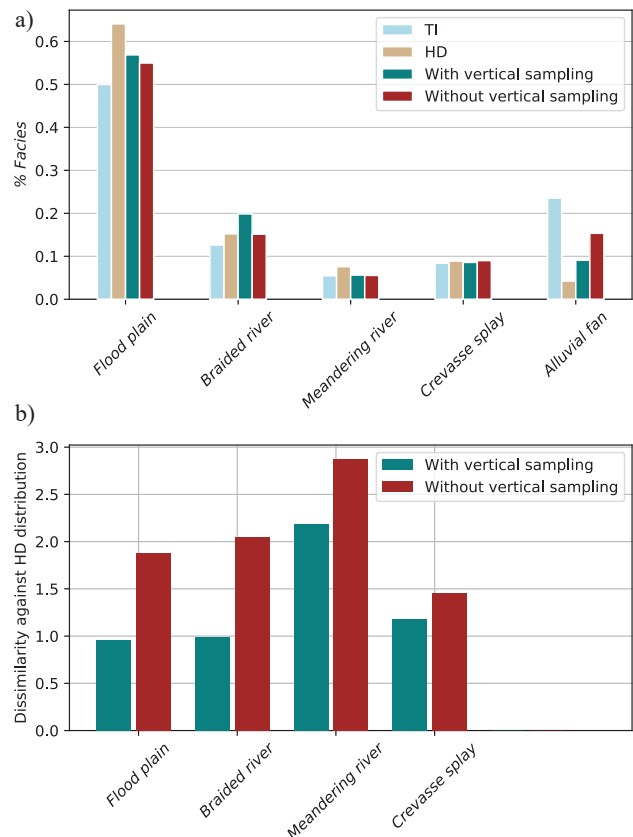

**Figure 10.** a) facies percentage in the training image, the hard data set, 3D simulation with vertical sampling and 3D simulation without vertical sampling. b) the dissimilarity values of the two simulation sets. The dissimilarity values are calculated against the vertical size distribution of the hard conditioning data.

Overall, the simulation set using the vertical sampling strategy possesses distributions closer to the conditioning data (smaller dissimilarities value) and produces vertically connected objects. The set using the vertical sampling strategy is composed of a larger number of thick objects as compared to the simulations set not using the sampling approach. Figure 10b and Fig. 11b show the beneficial impact of the vertical sampling approach on the simulation outputs.

## 5   Discussion and Conclusion

This study proposes a new workflow for the simulation of complex heterogeneous aquifers. Unlike more classical MPS studies, which rely on large primary or secondary hard data sets such as geophysics (Strebelle et al., 2002; Barfod et al., 2018; Høyer et al., 2017), this work relies on conceptual knowledge and auxiliary information.

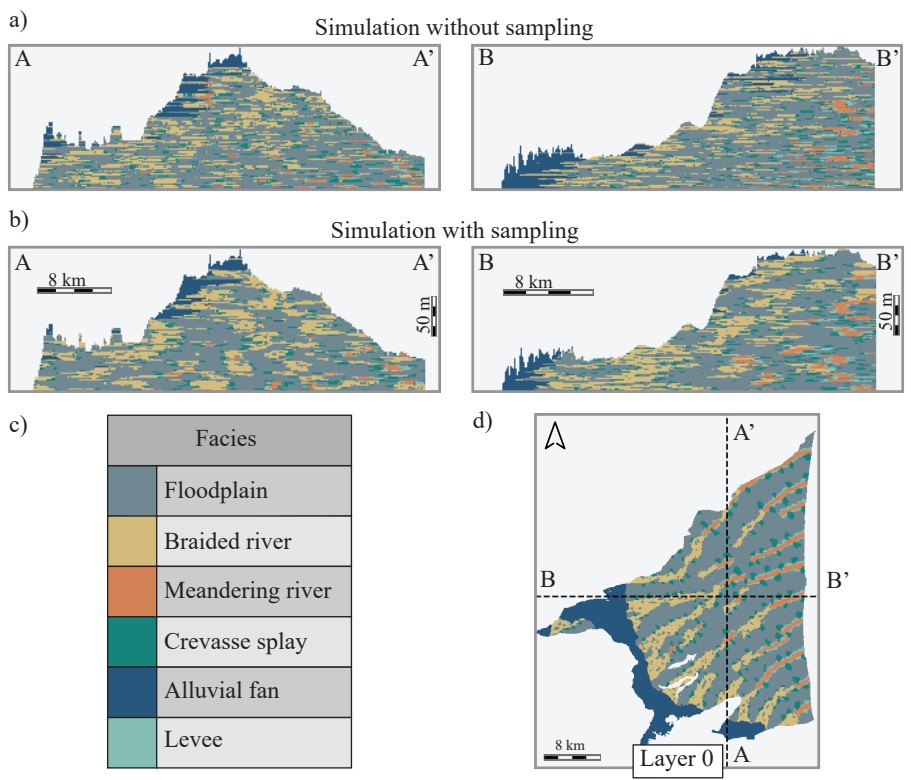

**Figure 11.** Cross-sections through two simulations, presented in the transformed space. a) two perpendicular vertical cross-sections through a simulation generated without the sampling approach. b) the same cross-sections through a simulation generated with the sampling approach. The braided and meandering facies display more vertical connection. c) the six simulated facies. d) Map view of the bottom layer of the simulation indicating the locations of the cross sections.

The main novelties within this workflow are the use of 2D simulations accounting for trends computed by solving a diffusion equation, the use of two continuous maps of rotation angles to account for the uncertainty on the paleo-orientations, and the

465 transfer of conditioning data from layer to layer to constrain the vertical transitions between the facies.

Solving numerically the diffusion equation allows to account easily for the complex geometry of the extension of the sedimentary basin when computing the trend map. Using that technique, it is straightforward to impose prescribed values of the trend on certain parts of the boundary and to ensure that the gradient of the trend will remain perpendicular to the sides of the domain.

The proposed approach is simpler and faster than one based on 3D TI. It has the advantage of being more flexible during the development of the model, where all of the elements can be easily adapted to the specific case and tested. In particular, the possibility of using auxiliary information allows the modeller to approach each problem with a different angle making MPS, and especially DeeSse very flexible.

The study of the Roussillon plain shows the importance of testing different TIs to obtain acceptable structures. The TI must be created to reflect the general geological knowledge available for the study site and it must respect the interpreted data. One of the strengths of the MPS method is that it allows to test rapidly different concepts that can be discussed and adapted. Moreover, the use of complex auxiliary variable and continuous rotation maps allow the final model to account for that information while honoring the borehole data.

The robustness of the proposed methodology has been tested with two sets of 50 simulations, one set with the vertical sampling approach and the other one without it. Despite its relatively simple method, the vertical sampling improves the vertical object size reproduction. The simulation without vertical sampling misses to create the large vertically connected objects, where the simulations with vertical sampling have a distribution more comparable to the hard data set distribution (Fig. 10b). These improvements, regarding the vertical objects size reproduction, are important, since the meandering river deposits and the braided river deposits are known to have high aquifer potentials. Recreating the vertical connectivity of these objects is a key parameter for the future use of the geological model for the hydro-characterization of the aquifer.

The probability and the entropy maps show that the important sedimentary concepts have been well integrated into the model. The facies proportion distributions of the different objects are satisfactorily reproduced and are constrained by both the boreholes facies distribution and the TI distribution. The probability and the entropy maps also show a lack of hard data. Indeed, the hard data set used may not be fully representative of the facies distribution of the Pliocene and the simulation can suffer from this bias. Regarding the vertical sampling approach, even if it improves the realism of the simulated object, the simulated shapes would benefit from additional constraints. The vertical transition matrix inferred from the boreholes can also present a bias due to their location or their non-representativeness of the real transition matrix.

Therefore, the model of the Roussillon plain would clearly benefit from additional boreholes with gamma ray and resistivity logs at locations that are not yet constrained. A denser data set would permit to conduct a meaningful cross validation exercise as suggested by Juda et al. (2020). At present, the data are not sufficient to really test carefully the predictive power of the MPS model. The model of the Roussillon plain would also benefit from additional information regarding the geometry of the sedimentological objects (their width, their lateral distribution...). Analogue data or geophysical inputs could help to better understand these geometries and could help in characterizing the transition zone between braided and meandering river deposits.

Finally, despite these possible improvements, this work demonstrates the applicability of DeeSse and of the proposed workflow to simulate complex internal aquifer heterogeneity at a regional scale.

*Author contributions.* AUTHOR CONTRIBUTION

V. Dall'Alba developed and coded the workflow, interpreted the geophysical logs, discussed the conceptual geological model, conducted all the numerical experiments, produced the figures and wrote the article. C. Duvail and B. Issautier developed the conceptual geological model and helped for the geophysical logs interpretation. J. Straubhaar and P. Renard supervised the research and in particular the design of the workflow and provided guidance for the MPS simulations. Y. Caballero coordinated

the whole project, and contributed to the hydrogeology section of this paper. Finally, all authors have helped during the writing process of this paper.

*Competing interests.*   COMPETING INTERESTS

The authors declare that they have no conflict of interest.

*Acknowledgements.*   The authors thank the partners of the Dem'Eaux Roussillon project and the University of Neuchâtel for supporting the development of this work.

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
