# Peer review of "3D Multiple-point Statistics Simulations of the Roussillon Continental Pliocene Aquifer using DeeSse"

_Hydrology and Earth System Sciences, 2020_

## Referee Comment (RC1) · Anonymous Referee #1 · 24 May 2020

Review of the manuscript hess-2020-96
"3D Multiple-point Statistics Simulations of the Roussillon
Continental Pliocene Aquifer using DeeSse"
by Dall'Alba V, Renard Ph, Straubhaar J, Issautier B, Duvail C,
Caballero Y

25 May 2020

**1 General comments**

This manuscript provides an improvement of the MPS implementation through the direct sampling algorithm, in order to design a method for the reconstruction of aquifer heterogeneity at scale lengths of tens of kilometers.

Overall, the work is interesting and deserves publication. The paper is generally well organized and written, but it can be improved following the suggestions given in the specific comments # 1 and 6. Some other weak scientific flaws can be fixed with a moderate to major revision.

**2 Specific comments**

1. The abstract is a long summary of the work, but it does not give a precise and clear image of the innovative content of the work. I think that it should be shortened and focused in a more appropriate way. Moreover, a similar comment applies to the introduction, which describes general properties of MPS, but does not properly introduce the specific methodological question which is faced with this work. The description given at lines 70 to 75 is not very exciting and informative. In my opinion, most of the material in section "3.1 Overview" should be anticipated in the introduction, in order to give a better presentation of the innovative character of this work at the very beginning of the paper.

2. For a long part of the manuscript, it was not clear to me whether the TIs were horizontal maps or vertical cross-sections. Moreover, the way in which 2D horizontal maps are used along the vertical direction should be better analysed. For instance, it would be useful to draw some vertical cross-sections in order to show the effects of the two simulation sets (with and without vertical sampling). In fact the analysis shown in figure 10 is not clear enough.

3. Section "3.2 Hard data set" could be improved.

   (a) There is some confusion between electrofacies and sedimentary facies.

   (b) And what about hydrofacies, which are ultimately the most important for hydraulic conductivity?

   (c) What about lithological logs? Usually they are available if a borehole is drilled for geophysical logs.

   (d) Details about the data set, e.g., position and borehole depth, are missing.

4. Section "3.3 Training images" is not very convincing. It shows that different TIs give different results and some of these are not appropriate with the geological structure of the study area. This is well known and was clearly proved by some of the authors in previous papers. It is well known that the TI should mimic the structures which are expected in the study area and this should be known *a priori* from geological studies. Moreover, further details should be given for figure 2a, which shows a strange sedimentary structure (see specific comment # 14). I am afraid that the term "braided river facies" is probably used in a non rigorous way. In fact, figure 2b shows the typical structure of a "braided river", with a great number of intersecting channels. In other words, areas characterised by meandering rivers show a very strong heterogeneity at relatively fine scale. This is not properly represented by the TIs.

5. Figure 8 shows that high probability for "floodplain" facies determines approximately linear structures. It seems that these structures separate the similar geometrical features observed for "braided river" and "meandering

river" facies. Is this right? This seems to be implicitly stated also in the text. In individual simulations, "floodplain" facies should be more widely distributed, shouldn't it? Why these maps show a different structure? Is this due to the constraint given by the elongated features for highest probability of "river" facies?

6. The orientation is missing in all the figures and the scale length is missing in almost all the figures.

**3  Technical comments**

1. Line 28. The acronym "PC" is used for the Continental Pliocene aquifer. Moreover, in a couple of sentences, I was confused and I read PC as "personal computer". I understand that "PC" is probably the correct acronym based on initials of French words, but I think that "CP" would be more appropriate as an acronym for the English name.

2. Line 40. Correct "1974)".

3. Line 58. Correct "Hu, 2008)".

4. Line 68. Substitute "," with ".".

5. Lines 95 to 97. I recommend the authors to carefully follow the international recommendations on the use of SI units and style conventions, in particular the guideline # 12 at the following URL:
   `https://physics.nist.gov/cuu/Units/checklist.html`.
   This applies also to other parts of the manuscript.

6. Lines 95, 97, 110. Substitute "extend" with "extent" or "extension".

7. Lines 99, 199, 213, 215, 217, 223-225, 286, 344, 345, 361, 362, 404, 419. I think the use of "meander" as adjective is not correct. I suggest to substitute "meander river" with "meandering river".

8. Line 104. Substitute "plain itself" with "floodplain".

9. Line 112. Substitute "in" with "at" before "some locations". Rephrase "up to 8m higher on average".

10. Line 117. Substitute "of" with "by".

11. Line 166. It is not clear if the TI is a map in the horizontal plane or a vertical cross section.

12. Line 171. Which is the direction of the $x$ coordinate axis?

13. Lines 196-197. Clarify the expression "By studying the evolution of these response curves".

14. Line 212. The expression "an analogue river system from northern Italy" does not provide a useful information. Which river? Which kind of geological setting? Moreover, from figure 2a, the braided river facies cover an extended area and does not properly represent the internal heterogeneity of a braided river system.

15. Line 216. Substitute "meander objects" with "meanders".

16. Lines 285-286. Why "the best way to control the vertical continuity was to sample only from three facies"? Can you comment on this and explain this result?

17. Line 319. Substitute "doesn't" with "does not".

18. Line 340. Substitute "are" with "is".

19. Line 385. Add "a" before "complex".

20. Lines 401 to 403. This remark is not so evident from the analysis of the results.

21. Line 495. Erase "Tectonophysics".

22. Figure 3. Substitute "c)" with "b)".

---

## Referee Comment (RC2) · Anonymous Referee #2 · 25 May 2020

1. The starting point for the proposed strategy, as stated in the abstract, is a conceptual model. The conceptual model is considered to be known deterministically. There is no mention of alternative, conceptual models. This is not trivial, and the authors need to justify this approach and suggest how to relax the constraint imposed by using a single conceptual model (see abstract lines 3-5)

2. The strategy presented here is smart in that it assimilates concepts from geology with geostatistical concepts. For example, the use of a physical-mathematical model for establishing the spatial evolution of sedimentary patterns. But some additional work is needed.

[Figure]

3. I recall some work done along this line (Item 2 above) by Steve Gorelick's group. As I recall, it requires using weather patterns over geological time scales, for boundary conditions. So, different patterns would have a strong effect on the patterns mentioned in (2). I am trying to figure out how climate/weather conditions permeated into the modeling of sedimentary patterns. There is a brief and incomplete description of the solution of the diffusivity equation. Just a couple of lines, between Line 170 and Line 175, on the topic. The authors need to shed more light on this aspect of their approach, and to show how the math/physical model used in support of (2) is actually constructed.

4. Line 65: Listing the advantages of the a 2010 simulation model, the authors state "... no probability is computed..". Question is: why is that an advantage? What are the positive and negative implications or avoiding probabilistic models? In Sections 4.2 and 4.3 there is a reference to probabilistic models. Confusing, and some clarification is required.

5. At the top of Section 4.2, the authors state as follows: "Simulating a large number of realizations enables us to calculate probability maps". That is obviously true: when you generate multiple realizations, you can compute probabilities. Question is: what is the connection between these probabilities, on one hand, and uncertainty and risk, on the other? The authors need to make a convincing case that they model uncertainty accurately. Without it, they can only say that they can generate images.

6. We need to see how the innovation (generating sedimentary patterns using a math/physical model) proposed in this study could make a difference. How would the generated images look like without the innovation? How does this innovation help in reducing uncertainty and improving accuracy? Some sort of cross-validation study (comparing results obtained with and without the improvement) could be helpful

---

## Referee Comment (RC3) · Anonymous Referee #3 · 17 Jun 2020

This paper presents a promising way of applying Direct Sampling, while avoiding the necessary step of creating a 3D Training Image. The presented results are, however, missing a presentation of the vertical connectivity of the simulations. This is an essential part of the presented method, where in order to replace the 3D TI with 2D TIs, vertical constraints are imposed. Currently the reader can not see how well these vertical constraints worked on the results.

Please also note the supplement to this comment: https://www.hydrol-earth-syst-sci-discuss.net/hess-2020-96/hess-2020-96-RC3-supplement.pdf

[Figure]

[Figure]

**Supplement:**

**Review: "3D Multiple-Point Statistics Simulations of the Rousiilon Continental Pliocene Aquifer using DeeSse"**

**General comments:**

This manuscript presents a method for large-scale 3D MPS simulations in areas without the necessity of creating a 3D Training Image, which can be a cumbersome and tedious task. Especially, the presented method focuses on areas with few geological observations and little to no geophysical data (soft conditioning data). Overall, a well written paper, albeit with some technical errors, which can easily be corrected. I recommend the paper for publication. Moderate review.

**Specific comments:**

- 1. Line 21-22: Explain why we want to skip this step. There are researchers who spend a large portion of their time building handheld 3D geological models and would not understand why it is an advantage to bypass the 3D TI.
- Line 51-61: A good overview, perhaps also mention Image Quilting Simulation (IQSIM) (Hoffimann 2017 - Stochastic simulation by image quilting of process-based geological models) and other more recent methods if any.
- 3. Line 68-69: Slightly elaborate "many options", so that the reader can grasp the advantages/disadvantages of DeeSse better.
- 4. Line 124-125: I think you mean two-point statistics. A variogram is not a geostatistical method, but a mathematical/statistical construct that is used in different geostatistical methods.
- Line 180-181: Making 3D Tis is not "too complex". In fact, in the literature, many studies are presented where handheld 3D geological models are created, which are essentially 3D TIs. Rephrase it to emphasize that it is a difficult, time consuming and subjective way of modeling, but NOT "too complex".
- 6. Line 201: What geological map? You should show it or cite it. If it is the one in Figure 1, then include a reference to figure 1 here.
- 7. Line 204-205: If it is essential to condition the model to the geological map, then you should describe how you do this.
- 8. Figure 3 caption: What is c) referring to? And b) is not mentioned. Needs to be fixed. Also, the grey model makes sense, but you should still describe it in the figure caption.
- Figure 4: In the b) and c) it would be a lot better if you added a thin black line to mark the outline of the layers, especially in relation to the stacked trend map in c). As for the current state of the figure I do not really get a lot of information from the stacked trend maps since all the colors of the different layers are identical by design.
- 10. Line256-257: What do the diffusivity equations look like? And what is considered "*proper boundary conditions*"?
- 11. Line 274: Insert a reference to Figure 5b) here
- 12. Line 276-277: Since river system are highly dynamic in nature, and since the reader has no idea about the sedimentation rates over the last 6 My in the given sedimentary basin, you should probably state why we can safely assume that the variation of the paleo orientations are encompassed by +/- 10 degrees of the values observed at the surface currently.

- 13. Line 285-287: Seems a bit discerning, can you elaborate as to why the vertical continuity was not as good when all 6 facies were included for sampling?
- 14. Figure 7: You should always introduce each sub figure in order a) -> b) -> c), not a) -> c) -> b). Also, a very important detail is that nowhere in the paper do you present a vertical slice, or profile, of the simulated models, so the reader cannot see how bad/well the vertical constraints worked in comparison to a full 3D TI. It would be easy to add a cross section view in this figure.
- 15. Line 346-348: How do the maps explain that the model is not over constrained? It might be obvious to you, but not necessarily the reader.
- 16. Lines 385-387: I think you are going need to touch on the strengths/weaknesses when making a comparison like that. The so-called *classical MPS studies* simply have a lot of geophysical/conditioning data available to them, and will, in large, be better conditioned to the actual subsurface. On the other hand, your method is clearly advantageous when you do not have a lot of geophysical/conditioning data available, but of course will not be as nicely conditioned to the actual subsurface. There are many places in the world where they need methods like this, since they do not have elaborate geophysical data sets available to them.
- 17. Line 408-409: How are they satisfactorily reproduced? Based on what? Is it solely based on the boreholes being conditioned correctly and the simulations resembling the TI? Then, it would be nice to show some statistics regarding how well the boreholes and simulations agree.

**Technical comments:**

- 1. In general, it is not called a "meander river", but a "meandering river". This needs to be fixed.
- 2. Line 28: The abbreviations for Marine Pliocene aquifer is PMS. Perhaps this is an abbreviation that makes sense in relation to the French name for the unit, but in order to make the paper more readable I recommend using MPA, which makes more sense.
- 3. Line 28: Similarly, the abbreviation for Continental Pliocene is PC. It would be more fitting to use CP.
- 4. Line 68: Change "It" to "it", *i.e.* no capital letters after comma.
- 5. Line 76-77: Add ":" after "The paper is structured as follows", followed by changing "The" to "the"
- 6. Line 78: Missing comma after DeeSse algorithm, and "Section 3" should not have a capital letter
- 7. Line 95: You mean the "extent" and not "extend".
- 8. Line 238: change "express" to "expressed"
- 9. Line 360: Change "the allivial fan dominate" to "the allivial fan dominates"

---

## Author Comment (AC1) · 26 Jun 2020

**3D Multiple-point Statistics Simulations of the Roussillon Continental Pliocene Aquifer using DeeSse (Valentin Dall'alba et al)**

**Anonymous Referee #1 (24 May 2020) :**

| General comments : | Responses : |
|---|---|
| This manuscript provides an improvement of the MPS implementation through the direct sampling algorithm, in order to design a method for the reconstruction of aquifer heterogeneity at scale lengths of tens of kilometers. | More precisely, the aim of the paper is to present a workflow allowing to apply the direct sampling technique to simulate aquifer heterogeneity at the regional scale. We do not improve the actual implementation of the MPS kernel. We employ the direct sampling MPS kernel into a global workflow. |
| Overall, the work is interesting and deserves publication. The paper is generally well organized and written, but it can be improved following the suggestions given in the specific comments # 1 and 6. Some other weak scientific flaws can be fixed with a moderate to major revision. | We are thankful to the reviewer for his/her overall evaluation and detailed editing of the paper. This is helping to improve the manuscript and clarify some aspects.

Below we discuss more precisely the different issues raised by the reviewer and how we plan to adjust the paper in consequence. |
| **Specific comments :** | **Responses :** |
| 1. The abstract is a long summary of the work, but it does not give a precise and clear image of the innovative content of the work.

I think that it should be shortened and focused in a more appropriate way. [...] | We agree that the abstract was long and maybe not sufficiently focused on the core of the paper. We propose a revised version that has been shortened. We tried to better highlight the novel aspects of the methodology. The proposed new abstract is the following :

"This study introduces a novel workflow to model the heterogeneity of complex aquifers using the multiple-point statistics algorithm DeeSse. We illustrate the approach by modeling the Continental Pliocene layer of the Roussillon's aquifer in the region of Perpignan (southern France). When few direct observations are available, statistical inference from field data is difficult if not impossible, and |

traditional geostatistical approaches cannot be applied directly. On the opposite, multiple-point statistics simulations can rely on a conceptual geological model provided using a training image. But since the spatial arrangement of geological structures is often non-stationary and complex there is a need for methods allowing to describe and account for the non-stationarity in a simple but efficient manner. The main aim of this paper is therefore to propose a workflow for these situations. The workflow is based on the direct sampling algorithm DeeSse. The conceptual model is provided by the geologist as a two-dimensional non-stationary training image (TI) in map view displaying the possible organization of the geological structures and their spatial evolution. To control the non-stationarity, a 3D trend map is obtained by solving numerically the diffusivity equation as a proxy to describe the spatial evolution of the sedimentary patterns, from the source of the sediments to the outlet of the system. A 3D continuous rotation map is estimated from inferred paleo-orientations of the fluvial system. Both trend and orientation maps are derived from geological insights gathered from outcrops and general knowledge of processes occurring in these types of sedimentary environments. Finally, the 3D model is obtained by stacking 2D simulations following the paleo-topography of the aquifer. The vertical facies transition between successive 2D simulations is controlled partly by the borehole data used for conditioning. But we also account for vertical probability of transitions derived from the borehole observations by simulating a set of conditional data points from one layer to the next. This process allows us to bypass the creation of a 3D training image which may be cumbersome in some situations while honoring the observed vertical continuity."

[...] Moreover, a similar comment applies to the introduction, which describes general properties of MPS, but does not properly introduce the specific methodological question which is faced with this work. The description given at lines 70 to 75 is not very exciting and informative. In my opinion, most of the material in section "3.1 Overview" should be anticipated in the introduction, in order to give a better presentation of the innovative character of this work at the very beginning of the paper.

Similar to the abstract, we propose to reconsider the text as suggested by the reviewer to provide a clearer outline of the approach in the introduction. We will try to emphasize more clearly the novel aspects of the approach.

As suggested by the reviewer we also proposed to add some more information about the motivation of the approach and an overview of the workflow in the introduction. The revised text will be placed after the presentation of the MPS methods and will be the following:

"The choice of a simulation technique to model an aquifer at a regional scale depends from different factors. One important aspect is the amount of data available. When the amount of data is large, it is possible to infer rather accurately the statistics describing the spatial variability from the data. Probability distributions about the different rock types, variograms, and spatial trends can be directly estimated and used in the simulation process. This situation often occurs in the mining industry for example where very large number of drill holes are made during the exploitation of an ore deposit. The situation is very different in other situations, such as the Roussillon plain, where only a few boreholes are available for a large study area. It becomes then difficult if not impossible to estimate accurately those statistical parameters from the data set. One has then to rely more heavily on indirect data, geological concepts, and analogy with other sites. In these situations, one could borrow statistical distributions, variograms and orders of magnitude of correlations lengths from data bases of similar environments such as those developed by \citet{colombera2012database}. The issue with that approach is that the simulations may be constrained only by a few data points and therefore the final variability among the simulations will be excessively large and the geological features will not be properly represented because the field data will not compensate the lack of geological concept in a variogram based geostatistical approach. An object based method would better respect the geological knowledge because the user will have to explicitly define the shape of the

objects, and this approach could be an interesting solution for these situations with an important data gap. But here, we rather consider the use of MPS. As for the object based approach, it allows integrating directly geological knowledge in the stochastic simulation process.

Another very important aspect to take into account at the regional scale are the statistical non-stationarities resulting from geological processes such as the location of the sources of the sediments, their transport, deposition, and so on. The application of MPS to a real case requires therefore more than just an efficient MPS code and a good training image. It requires also to develop a methodology and a workflow to account for all those aspects.

The aim of this paper is therefore to introduce such a global workflow allowing to incorporate most of the available geological knowledge into a plausible heterogeneity model and to illustrate the method on the Roussillon plain. The workflow is generic and can be applied to any other case where the available data are scarce compared to the geological knowledge. The workflow includes a series of steps that are described in detail in the paper. Based on the borehole and geological knowledge of the site, a plan view non-stationary training image displaying the main sedimentological features is designed. In this paper, we limit ourselves to the construction of a 2D training image since there are many situations in which the cross sectional view at the scale of the aquifer is much less well known than the expected spatial organization of the sedimentary layers on a 2D horizontal plane. The vertical transitions are controlled using probability of transitions derived from the boreholes. To control the lateral transitions and non stationarity, 3D auxiliary maps representing a proxy of the evolution of the system from the source of the sediment to the output are modeled by solving a diffusivity equation. The boundary conditions imposed to the diffusivity equation allow to account for the paleo-input zones and the lateral geometry of the aquifer. In addition, the proposed workflow accounts for the paleo orientations of the sedimentary system

| | and its related uncertainty as inferred from field observations. The paper shows that such an approach can be efficient to simulate realistic alluvial systems matching the conceptual knowledge of the system." |
|---|---|
| 2. For a long part of the manuscript, it was not clear to me whether the TIs were horizontal maps or vertical cross-sections. | The paper already indicates, for example in lines 168-169, that the training image represents the expected pattern in a layer of the same age of deposition. We thought that this was clear enough. But to make the point clearer, we plan to provide the information earlier in the abstract and in the text to ensure that this is clear from the beginning.

We plan to correct the legend of figure 2 to clarify the TI orientation :
"**Horizontal TIs** associated with their corresponding 2D MPS simulations."

And modify lines 166-167 to clarify the fact that the TI is a horizontal representation of the system :
"The TI used for the Continental Pliocene layer is a 2D non-stationary **conceptual plan view of an alluvial system** composed of 6 sedimentary facies." |
| Moreover, the way in which 2D horizontal maps are used along the vertical direction should be better analysed. For instance, it would be useful to draw some vertical cross-sections in order to show the effects of the two simulation sets (with and without vertical sampling). | We agree with the reviewer and propose to introduce a new figure representing cross-sections of simulations with and without the sampling approach in order to effectively visualize its effect. This new figure 11 is presented at the end of this document. We also add a small description in section 4.4 :

"The impact of the sampling approach can also be easily observed when studying vertical cross-sections along the x and y axis in the transformed grid space (Fig. 11). In Fig. 11a the channels created by the stacking of the braided/meandering facies are vertically disconnected from each other. The impact of the sampling approach that leads to the creation of vertically connected objects can be observed in Fig. 11b. By opposition to the simulation that is not using the sampling strategy, it is now possible to observe "channels like" |

| | cross-sections in the simulation output." |
|---|---|
| In fact the analysis shown in figure 10 is not clear enough. | We propose to add some information on the dissimilarity index beginning at line 373 :

 "To quantify the impact of the vertical sampling strategy, following previous authors, we compared the distributions of the vertical runs (Mood, 1940, Boisvert et al 2007). To compute this indicator, the 3D grid is decomposed as a set of vertical columns of voxels. A vertical run is then defined as the length of a succession of the same facies values preceded and succeeded by a different facies. By computing the run length on all the columns for a given facies, one can compute the empirical distribution of runs for this facies. The same operation is conducted for all the facies. In addition, these empirical distributions are also computed on the borehole data. We then compute dissimilarity indices between the simulated and observed distributions for all the facies using a normalized euclidean distance. The closest to zero the dissimilarity value is, the more identical the distributions are and reciprocally (Fig. 10b). The alluvial fan facies is here not represented, because it is under-represented in the hard data set and a reference distribution cannot be inferred from it." |
| 3. Section "3.2 Hard data set" could be improved.
 (a) There is some confusion between electrofacies and sedimentary facies. | We understand the confusion between the two terms. We propose to modify the paper in order to use only the term **sedimentary facies**, which is more appropriate. |
| (b) And what about hydrofacies, which are ultimately the most important for hydraulic conductivity? | The reviewer raises an interesting remark. The main answer to this question is that two sedimentological units may have similar hydraulic conductivities but very different geometrical shapes. If we associate them to the same hydrofacies during the geostatistical simulation procedure, experience shows that the geometrical shapes of the two sedimentological units gets mixed and looses consistency. This is why, it is often better to model first the sedimentological units and then fill them with hydraulic properties. |

| | |
|---|---|
| | In addition, this process allows to account better for the hard data descriptions and the indirect geological knowledge and sedimental history of the area. |
| (c) What about lithological logs? Usually they are available if a borehole is drilled for geophysical logs. | The lithological logs are indeed available and used for the sedimentary facies description. We proposed to clarify this point in section 3.2 :

 "Hard data correspond to field observations assigned to cell values in the simulation grid. The hard conditioning data set of the Pliocene model is composed of 52 well logs (**lithological**, gamma-ray, and resistivity logs)... By studying the evolution of these response curves coupling with the study of the associated **lithological log**, the sedimentary facies…". |
| (d) Details about the data set, e.g., position and borehole depth, are missing. | The position of the boreholes was indeed missing. This has been corrected and included in figure 1. We also propose to add some information in the text regarding the borehole data, section 3.2 :

 "The boreholes are not homogeneously distributed on the plain but are mainly located along the Têt river and in the central zone of the Roussillon's plain (Fig. 1). Their depth range from 20 to 150 m and they are on average 77 m deep." |
| 4. Section "3.3 Training images" is not very convincing. It shows that different TIs give different results and some of these are not appropriate with the geological structure of the study area. This is well known and was clearly proved by some of the authors in previous papers. It is well known that the TI should mimic the structures which are expected in the study area and this should be known a priori from geological studies. | The reviewer is correct in stating that the comparison of several TIs is not novel and should be a rather standard step in any MPS simulation study. We do not claim that this is new.

 But we think that it is important to discuss that aspect in the framework of non stationary TI and simulations. The simulated patterns are often difficult to predict from the TI alone. Previous publications show that complex simulations can be obtained with very simple training images if proper parametrizations and trends are provided. The simulations will be very different from the TI and therefore the selection of the TI requires some trial and error |

| | testing. |
|---|---|
| | Therefore, we would like to keep that part which is important for the whole procedure in our opinion. This is a step in which several conceptual models can be tested. A point also mentioned by the reviewer and we agree with him on that. Testing various TI is a step of the workflow. We think that at least to illustrate these ideas, this part of the paper can be useful for some readers and should be kept. |
| Moreover, further details should be given for figure 2a, which shows a strange sedimentary structure (see specific comment# 14). | As presented in the last comment, all of the three TIs described an alluvial system represented with the same main spatial pattern evolution. The different shapes express within a facies through the three TIs are proposed in order to test and represent at different scales different sedimentological hypotheses (eg: whether the braided river deposit cut or not the alluvial fan facies).

The facies name must not be taken in a narrow sense as they represent more a location of the depositional environment within the whole alluvial system than a facies description. |
| I am afraid that the term "braided river facies" is probably used in a non rigorous way. In fact, figure 2b shows the typical structure of a "braided river", with a great number of intersecting channels. | We disagree with the reviewer but we did not explain clearly enough the reasoning behind this figure. The text will be corrected to better explain the argument.

As explained in the previous answer, it is important to consider different pattern configurations in different training images and check how this will be transferred to the regional simulations when combined with non stationarity parameters and trends. This idea led to the creation of three TIs and three different representations of the braided river facies.

In the first TI, the braided river facies represents the entire braided channel belt without describing its internal heterogeneity. In the second and third TI, some internal heterogeneity is included in the concept.

The reason why the entire braided river channel belt can be considered as one single channel is |

| | that it happens that there is little potential to preserve low permeability sediments in the braided river channel belt. There is internal heterogeneity in the braided system for sure at a meter scale. But at the reservoir scale, one may consider that the important contrast is the one between the braided river belt and the floodplain. Therefore, it could be reasonable to model the system in that manner. |
|---|---|
| In other words, areas characterised by meandering rivers show a very strong heterogeneity at relatively fine-scale. This is not properly represented by the TIs. | The same explanation stands for the meandering river, where we decided after detailed discussions with the geologists to represent the meandering river belt and not each individual meanders. This approach is explained by figure 2 e) and is described in detail in the PhD thesis of *Issautier, Benoît. (2011). Impact des hétérogénéités sédimentaires sur le stockage géologique du CO2. University of Aix-Marseille, France : https://www.theses.fr/2011AIX10136* |
| 5. Figure 8 shows that high probability for "floodplain" facies determines approximately linear structures. It seems that these structures separate the similar geometrical features observed for "braided river" and "meandering river" facies. Is this right? This seems to be implicitly stated also in the text. | The geological concept and training images imply indeed an alternation of channels (either meandering or braided) and flood plain. When a hard data indicates the presence of one of the facies, it will impose a high probability of occurrence for this facies at the hard data location, but also upstream and downstream since the channel belts have this rather linear structure. The shape will not be exactly linear because a tolerance is used for the rotation of the channel belts in the plain and the distance between the channels is not constant in the training image. Once a facies is placed, the geological consistency implies that at some lateral distance the other facies (flood plain or channel belt) will have to be present. Therefore, the general pattern identified by the reviewer is correct but the situation is slightly more complex than simple linear trends. We propose to update the text to make the point as clear as possible in the paper. |
| In individual simulations, "floodplain" facies should be more widely distributed, shouldn't it? Why these maps show a different structure? Is this due to the constraint given by the elongated | In the individual simulations (for example Fig 7), the flood plain is rather widely distributed. The spacing observed between the river belt in the simulation output (figure 7) corresponds to |

| | |
|---|---|
| features for highest probability of "river" facies? | the indications provided by the geologists on the site. In addition, the overall proportion of flood plain is clearly larger than the channels on every single realization. This is visible in figure 7. In the ensemble of simulations and on the probability maps (figure 8) the flood plain facies has the highest probability of occurrence as compared to the other facies. Therefore, we do not think that the flood plain facies should be more widely distributed. |
| 6. The orientation is missing in all the figures and the scale length is missing in almost all the figures. | Yes we agree. The figures will be modified in order to add scale length and orientation. |
| **Technical comments :** | **Responses :** |
| 1. Line 28. The acronym "PC" is used for the Continental Pliocene aquifer. Moreover, in a couple of sentences, I was confused and I read PC as "personal computer". I understand that "PC" is probably the correct acronym based on initials of French words, but I think that "CP" would be more appropriate as an acronym for the English name. | We understand the possible confusion for the reader, however, this acronym is used by all of the persons that are working in the area. We prefer to keep it for consistency. But we will replace the acronym as much as possible in the revised version of the paper and propose to use the term "Pliocene" when referring to the "Continental Pliocene layer". We will introduce the terminology at the end of the Geology subsection 2.1 :

 "In the following, and because we do not consider the deeper Marine Pliocene formations in this paper, we refer to the Continental Pliocene layer and aquifer (usually denoted PC in the area) as Pliocene." |
| 2. Line 40. Correct "1974)". | Agree. We will correct this point. |
| 3. Line 58. Correct "Hu, 2008)". | Agree. We will correct this point. |
| 4. Line 68. Substitute "," with ".". | Agree. We will correct this point. |
| 5. Lines 95 to 97. I recommend the authors to carefully follow the international recommendations on the use of SI units and style conventions, in particular the guideline # 12 at the following URL: https://physics.nist.gov/cuu/Units/checklist.htm. | Agree. We will correct this point. |

| | |
|---|---|
| This applies also to other parts of the manuscript. | |
| 6. Lines 95, 97, 110. Substitute "extend" with "extent" or "extension". | Agree. We will correct this point. |
| 7. Lines 99, 199, 213, 215, 217, 223-225, 286, 344, 345, 361, 362, 404, 419. I think the use of "meander" as adjective is not correct. I suggest to substitute "meander river" with "meandering river". | Agree. We will correct this point. |
| 8. Line 104. Substitute "plain itself" with "floodplain". | Agree. We will correct this point. |
| 9. Line 112. Substitute "in" with "at" before "some locations". Rephrase "up to 8m higher on average". | We propose to change the sentence to :

 "In the 1960s, the piezometric level was on average 8 m higher as compared to the 2012 data and even artesian at some locations." |
| 10. Line 117. Substitute "of" with "by". | Agree. We will correct this point. |
| 11. Line 166. It is not clear if the TI is a map in the horizontal plane or a vertical cross-section. | We change it to :

 "The TI used for the Pliocene is a 2D non-stationary conceptual plan view of an alluvial system composed of 6 sedimentary facies." |
| 12. Line 171. Which is the direction of the x coordinate axis? | The TI is not spatially oriented, however, the x-direction can be assimilated to the east-west direction on the grid. |
| 13. Lines 196-197. Clarify the expression "By studying the evolution of these response curves". | We propose to reformulate the expression as follow :

 "By analyzing the gamma-ray and resistivity responses at a certain depth coupling with their vertical evolution, it is possible to identify and assigned a sedimentary facies to a certain depth range." |
| 14. Line 212. The expression "an analogue river system from northern Italy" does not provide a | We agree with the reviewer that the expression "an analogue river system" can be misleading and lack precision. We used satellite images of |

| | |
|---|---|
| useful information. Which river? Which kind of geological setting? Moreover, from figure 2a, the braided river facies cover an extended area and does not properly represent the internal heterogeneity of a braided river system. | the Tagliamento river, which is located in Northern Italy near the town of Udine and close to the Slovenia border to create the first TI. We propose to change the description line 212 in the new version of the article :

"The first TI (Fig. 2a) is created based on visual interpretation of satellite images of the Tagliamento river, which is located in Northern Italy near the town of Udine and close to the Slovenia border. The entire channel belt is considered as the deposition zone. Moreover, this TI neither represents the small scale internal structures of the river deposits nor the levee structures."

As for the comment on the "braided river facies", this point is already answered above within the specific comment 4. |
| 15. Line 216. Substitute "meander objects" with "meanders". | Agree. We will correct this point. |
| 16. Lines 285-286. Why "the best way to control the vertical continuity was to sample only from three facies"? Can you comment on this and explain this result? | Since the "floodplain facies" is the most frequent, sampling the facies at random location leads to an over-representation of the flood plain and tends to bias the MPS simulations.

After some tests, it appeared that the easiest way to control the connectivity of the objects of interest was to sample only those facies (alluvial fan, braided and meandering river).

We also decided to not sample the levee and crevasse splay facies in order to avoid constraining the whole structure of the fluvial objects too heavily.

We propose to add these explanations in the revised version of the manuscript. |
| 17. Line 319. Substitute "doesn't" with "does not". | Agree. We will correct this point. |
| 18. Line 340. Substitute "are" with "is". | Agree. We will correct this point. |
| 19. Line 385. Add "a" before "complex". | Agree, we change the end of the sentence to the plural form :
"...for the simulation of complex heterogeneous |

| | |
|---|---|
| | aquifers." |
| 20. Lines 401 to 403. This remark is not so evident from the analysis of the results. | We proposed to clarify the explanations of figure 10 in section 4.4 and cite this figure in lines 401-403. Moreover, a new figure presenting cross-sections through the simulation with and without the sampling approach will be introduced in the revised version of the paper. The figure will clarify the effect of the sampling approach on the vertical connectivity of the river beds. |
| 21. Line 495. Erase "Tectonophysics". | Agree. We will correct this point. |
| 22. Figure 3. Substitute "c)" with "b)". | Agree. We will correct this point. |

**New figure 11 :**

[Figure]

a) Simulation without sampling

b) Simulation with sampling

c)

| Facies | |
|---|---|
| | Floodplain |
| | Braided river |
| | Meandering river |
| | Crevasse splay |
| | Alluvial fan |
| | Levee |

d)

---

## Author Comment (AC3) · 26 Jun 2020

**3D Multiple-point Statistics Simulations of the Roussillon Continental Pliocene Aquifer using DeeSse (Valentin Dall'alba et al)**

**Anonymous Referee #3 (17 June 2020) :**

| General comments : | |
|---|---|
| This manuscript presents a method for large-scale 3D MPS simulations in areas without the necessity of creating a 3D Training Image, which can be a cumbersome and tedious task. Especially, the presented method focuses on areas with few geological observations and little to no geophysical data (soft conditioning data).
Overall, a well written paper, albeit with some technical errors, which can easily be corrected. I recommend the paper for publication. Moderate review. | We are thankful to the reviewer for his/her overall evaluation and its detailed comments on the paper. The specific and technical comments will help to improve the manuscript.

Below we discuss more precisely the different issues raised by the reviewer and how we plan to adjust the paper in consequence. |
| **Specific comments :** | |
| 1- Line 21-22: Explain why we want to skip this step. There are researchers who spend a large portion of their time building handheld 3D geological models and would not understand why it is an advantage to bypass the 3D TI. | This is an important point that was not sufficiently well explained in the introduction. We propose to add a paragraph explaining why we think that this is an important aspect of the methodology :

"When using MPS, an important question is the construction of the training image. We first want to note that the conceptual sedimentological models are usually represented in 2D map views or block diagrams and geologists are used to express their understanding of a system by drawing such maps and cross sections. Furthermore, remote sensing data or geological maps are widely available and can be used to refine these 2D conceptual models. Accessing 2D training images is therefore easy and simple. However, the standard MPS workflow requires a 3D training image to generate 3D simulations. Getting the 3D training image from 2D concepts is not a simple task. It may require a significant |

| | |
|---|---|
| | amount of tedious work to construct manually a 3D training image from the 2D concepts. Therefore, previous research was devoted to the design of MPS algorithms able to use 2D training images directly as input for 3D simulations (Comunian et al., 2012; Cordua et al., 2016). Here, we propose a simple approach that allows the user to avoid the step of the 3D training image construction. This is not mandatory. If a 3D training image is available, it can easily be used in the workflow, but if it is not available it should not be a limitation as we will illustrate in the paper." |
| 2- Line 51-61: A good overview, perhaps also mention Image Quilting Simulation (IQSIM) (Hoffimann 2017 - Stochastic simulation by image quilting of process-based geological models) and other more recent methods if any. | We will add this reference in the revised version of the manuscript, we also add more recent advances made on the DeeSse algorithm : "In other algorithms, such as FILTERSIM (Zhang et al., 2006), CCSIM (Tahmasebi et al., 2012) or IQSIM (Hoffimann et al., 2017), the simulation grid is filled by directly pasting or quilting patches, i.e. several pixels at a time. FILTERSIM uses a set of filters to reduce the dimension of the problem, whereas CCISM is based on cross-correlation between patches. The image quilting simulation (IQSIM) proposes a new approach that bypasses traditional ad-hoc weighting of auxiliary variables." |
| 3- Line 68-69: Slightly elaborate "many options", so that the reader can grasp the advantages/disadvantages of DeeSse better. | We proposed to add recent publication on the code DeeSse but would prefer to no explain/list in details all of the DeeSse options and parameters : "More details about the features of the DeeSse code are provided in Meerschman et al. (2013); Straubhaar et al. (2016, 2020). " |
| 4- Line 124-125: I think you mean two-point statistics. A variogram is not a geostatistical method, but a mathematical/statistical construct that is used in different geostatistical methods. | We agree on this point and will replace "variogram" with "two-points statistics" in line 124-125 of the manuscript. |

| | |
|---|---|
| 5- Line 180-181: Making 3D Tis is not "too complex". In fact, in the literature, many studies are presented where handheld 3D geological models are created, which are essentially 3D TIs. Rephrase it to emphasize that it is a difficult, time consuming and subjective way of modeling, but NOT "too complex" | The reviewer is right on the subjective use of "too complex" and that some other studies lead to the creation of 3D geological models, which could be used as TI. We propose to review the manuscript with this new version :

 "As discussed in the introduction, this approach allows avoiding the construction of a 3D TI that could be cumbersome." |
| 6- Line 201: What geological map? You should show it or cite it. If it is the one in Figure 1, then include a reference to figure 1 here. | We will add the reference to the cited geological map. |
| 7- Line 204-205: If it is essential to condition the model to the geological map, then you should describe how you do this. | We agree on this point and proposed to add some information on the process :

 "The hard conditioning data set also incorporates geological information from the geological map of the Roussillon (Genna,2009). These data correspond to the mapped Pliocene alluvial fan outcrops. We transformed the polygons from the geological map toward conditioning data set for the simulation. The facies assigned to these outcrops corresponds to the alluvial fan facies." |
| 8- Figure 3 caption: What is c) referring to? And b) is not mentioned. Needs to be fixed. Also, the grey model makes sense, but you should still describe it in the figure caption. | It was a typo in the caption. We propose this new caption for the figure 3 :

 "a) 3D grid of the Pliocene, dark green (the grey volume representing the transformed space). b) transformed grid (flattened space) of the Pliocene layer inside which the 2D simulations are simulated, dark orange (the grey volume representing the original space). The vertical scale is exaggerated for this representation. View from the South of the area toward the North." |
| 9- Figure 4: In the b) and c) it would be a lot better if you added a thin black line to mark the outline of the layers, especially in relation to the stacked trend map in c). As for the current state of the figure I do not really get a lot of information from the stacked trend maps since all the colors of the different layers are identical by design. | The stacked trend map only shows the progradation of the trend as we move upward in the layers, the dark blue color is moving gradually towards the sea as the system evolves.

 Black contour lines will be added to the figure for a better presentation of the 3D view. |

| | |
|---|---|
| 10- Line 256-257: What do the diffusivity equations look like? And what is considered "proper boundary conditions"? | To answer this comment, we propose to rewrite the text as follows :

"In the flattened space grid, the auxiliary variables are computed by solving numerically a diffusivity equation in steady-state ($\Delta h=0$, with $\Delta$ representing the Laplacian operator) for each of the 2D layers composing the 3D grid. The problem is solved using a finite element mesh following the exact geometry of the domain. The boundary conditions are: prescribed values $h(x)=h_0$ on some parts of the boundary; and $\nabla h(x) \cdot n_x = 0$ on the rest, meaning that the gradient of $h(x)$ should be perpendicular to the vector $n_x$ that is normal to the boundary at that location, i.e. the maximum variation of the trend must be parallel to the boundary." |
| 11- Line 274: Insert a reference to Figure 5b) here | Agree, this will be corrected. |
| 12- Line 276-277: Since river system are highly dynamic in nature, and since the reader has no idea about the sedimentation rates over the last 6 My in the given sedimentary basin, you should probably state why we can safely assume that the variation of the paleo orientations are encompassed by +/- 10 degrees of the values observed at the surface currently. | It is true that river systems are highly dynamic and that their bed orientations can vary through time. However, since our TI encompasses the whole river bed, as shown in figure 2d, we do not expect to see strong orientation changes.

The rotation map is derived from interpretation and thus cannot be taken as true fixed values of the paleo orientations. The +/- 10 degrees of tolerance is fixed to take into account this uncertainty. This tolerance also helps to not overconstrain the model and to accommodate the location of the patterns to the hard data during simulations.

We think that the sentence "The orientation map is based on interpretation and therefore uncertain. DeeSse allows to account for this uncertainty. A tolerance of +/- 10° is considered and added/subtracted to the kriged map to obtain two rotation maps..." gives enough information on the reader on the fact that the kriged value is not the true orientation of the paleo river and that some flexibility must be left to the algorithm to simulate the orientation of the patterns. |

| | |
|---|---|
| 13- Line 285-287: Seems a bit discerning, can you elaborate as to why the vertical continuity was not as good when all 6 facies were included for sampling? | This question has already been raised by the reviewer#1. We propose to copy the answer of this comment :

Since the "floodplain facies" is the most frequent, sampling the facies at random location leads to an over-representation of the flood plain and tends to bias the MPS simulations.

After some tests, it appeared that the easiest way to control the connectivity of the objects of interest was to sample only those facies (alluvial fan, braided and meandering river).

We also decided to not sample the levee and crevasse splay facies in order to avoid constraining the whole structure of the fluvial objects too heavily.

We propose to add these explanations in the revised version of the manuscript. |
| 14- Figure 7: You should always introduce each sub figure in order a) -> b) -> c), not a) -> c) -> b). Also, a very important detail is that nowhere in the paper do you present a vertical slice, or profile, of the simulated models, so the reader cannot see how bad/well the vertical constraints worked in comparison to a full 3D TI. It would be easy to add a cross section view in this figure. | The caption for figure 7 has been corrected, the b) and c) were simply swapped.

We understand the need of cross section view in order to visually inspect the effect of the vertical sampling approach on the simulation outputs. We propose a new figure 11 including these elements. The figure is presented at the end of this document. We also add a small description in section 4.4 :

"The impact of the sampling approach can also be easily observed when studying vertical cross-sections along the x and y axis in the transformed grid space (Fig. 11). In Fig. 11a the channels created by the stacking of the braided/meandering facies are vertically disconnected from each other. The impact of the sampling approach that leads to the creation of vertically connected objects can be observed in Fig. 11b. By opposition to the simulation that not using the sampling strategy, it is now possible to observe "channels like" cross-sections in the simulation output." |

| | |
|---|---|
| 15- Line 346-348: How do the maps explain that the model is not over constrained? It might be obvious to you, but not necessarily the reader. | We agree with the reviewer that more information needs to be included for the reader. We propose a revised version of this sentence for the manuscript :

"These maps also highlight the fact that the model is not over-constrained by neither the TI nor the HD. Focusing on the meandering and braided river facies. We can observe that in some locations the probability maps show high probability values due to the presence of conditioning data. However, this probability decreases proportionally to the distance to the hard data, which demonstrates that the model is not over-constrained by the conditioning data. Moreover, even when a river bed location is constrained with a hard data, the associated spacing with other river bed is not fixed and can fluctuate through the simulation set, which indicates that the TI does not lock the locations of the pattern during a simulation." |
| 16- Lines 385-387: I think you are going need to touch on the strengths/weaknesses when making a comparison like that. The so-called classical MPS studies simply have a lot of geophysical/conditioning data available to them, and will, in large, be better conditioned to the actual subsurface. On the other hand, your method is clearly advantageous when you do not have a lot of geophysical/conditioning data available, but of course will not be as nicely conditioned to the actual subsurface. There are many places in the world where they need methods like this, since they do not have elaborate geophysical data sets available to them. | We think that the presented method should not be viewed in opposition to other methods where geophysical data are available. There is no strength in lacking soft information, the objects are not well described neither in their size nor their location and validation approach cannot be performed when lacking conditioning data.

The proposed workflow is only an alternative that tries to take into account most of the available conceptual knowledge. |
| 17- Line 408-409: How are they satisfactorily reproduced? Based on what? Is it solely based on the boreholes being conditioned correctly and the simulations resembling the TI? Then, it would be nice to show some statistics regarding how well the boreholes and simulations agree. | The facies proportions are satisfactorily reproduced base on the fact that they are similar to the borehole facies proportion distributions, excepted for the alluvial fan proportion that is under-represented in the hard data but compensated with the influence of the TI.

We believe that this is shown in figure 10 a and fully explain in the sub-section 4.4. |

| Technical comments : | |
|---|---|
| 1- In general, it is not called a "meander river", but a "meandering river". This needs to be fixed. | Agree. We will correct this point. |
| 2- Line 28: The abbreviations for Marine Pliocene aquifer is PMS. Perhaps this is an abbreviation that makes sense in relation to the French name for the unit, but in order to make the paper more readable I recommend using MPA, which makes more sense.

3- Line 28: Similarly, the abbreviation for Continental Pliocene is PC. It would be more fitting to use CP. | This comment was already mentioned by the reviewer#1. Here is the proposed answer :

We understand the possible confusion for the reader, however, this acronym is used by all of the persons that are working in the area. We prefer to keep it for consistency. But we will replace the acronym as much as possible in the revised version of the paper and propose to use the term "Pliocene" when referring to the "Continental Pliocene layer". We will introduce the terminology at the end of the Geology subsection 2.1 :

"In the following, and because we do not consider the deeper Marine Pliocene formations in this paper, we refer to the Continental Pliocene layer and aquifer (usually denoted PC in the area) as Pliocene." |
| 4- Line 68: Change "It" to "it", i.e. no capital letters after comma. | Agree. We will correct this point. |
| 5- Line 76-77: Add ":" after "The paper is structured as follows", followed by changing "The" to "the" | Agree. We will correct this point. |
| 6- Line 78: Missing comma after DeeSse algorithm, and "Section 3" should not have a capital letter | Agree. We will correct this point. |
| 7- Line 95: You mean the "extent" and not "extend". | Agree. We will correct this point. |
| 8- Line 238: change "express" to "expressed" | Agree. We will correct this point. |
| 9- Line 360: Change "the alluvial fan dominate" to "the alluvial fan dominates" | Agree. We will correct this point. |

**New figure 11 :**

[Figure]

a) Simulation without sampling

A        A'       B        B'

50 m

b) Simulation with sampling

A        A'       B        B'

50 m

c)

| Facies |
| --- |
| Floodplain |
| Braided river |
| Meandering river |
| Crevasse splay |
| Alluvial fan |
| Levee |

d)

A'

B        B'

8 km

Layer 0    A

---

## Author Response (AR1)

**3D Multiple-point Statistics Simulations of the Roussillon Continental Pliocene Aquifer using DeeSse (Valentin Dall'alba et al)**

**We appreciate the helpful comments of the reviewers and editor. Please find below in black font, the summary of how we have addressed each reviewer's comments (blue font) in the revised manuscript. The line numbers refer to the revised manuscript.**

**Answer to Referee #1 :**

**General comments :**

This manuscript provides an improvement of the MPS implementation through the direct sampling algorithm, in order to design a method for the reconstruction of aquifer heterogeneity at scale lengths of tens of kilometers.

More precisely, the aim of the paper is to present a workflow allowing to apply the direct sampling technique to simulate aquifer heterogeneity at the regional scale. We do not improve the actual implementation of the MPS kernel. We employ the direct sampling MPS kernel into a global workflow.

Overall, the work is interesting and deserves publication. The paper is generally well organized and written, but it can be improved following the suggestions given in the specific comments # 1 and 6. Some other weak scientific flaws can be fixed with a moderate to major revision.

We are thankful to the reviewer for his/her overall evaluation and detailed editing of the paper. This is helping to improve the manuscript and clarify some aspects. Below we discuss more precisely the different issues raised by the reviewer and how we have adjusted the paper in consequence.

**Specific comments :**

The abstract is a long summary of the work, but it does not give a precise and clear image of the innovative content of the work. I think that it should be shortened and focused in a more appropriate way.

We agree that the abstract was long and maybe not sufficiently focused on the core of the paper. We proposed a revised version that has been shortened. We tried to better highlight the novel aspects of the methodology in the revised version (L1-19).

*Moreover, a similar comment applies to the introduction, which describes general properties of MPS, but does not properly introduce the specific methodological question which is faced with this work. The description given at lines 70 to 75 is not very exciting and informative. In my opinion, most of the material in section "3.1 Overview" should be anticipated in the introduction, in order to give a better presentation of the innovative character of this work at the very beginning of the paper.*

Similar to the abstract, we proposed to reconsider the text in the introduction as suggested by the reviewer to provide a clearer outline of the approach. We tried to emphasize more clearly on the novel aspects of the approach. As suggested we also added some more information about the motivation of the approach and an small overview of the workflow in the introduction (L102-115). However, we decided to keep the Overview sub-section (L202-230) as we think it reintroduces the main steps of the workflow and helps the reader to wrap his/her mind around the different steps and the different elements composing the final model. The Overview sub-section has been re-write in order to be more clear.

*For a long part of the manuscript, it was not clear to me whether the TIs were horizontal maps or vertical cross-sections.*

We understand the possible confusion and explicitly defined that the TIs are horizontal maps view in the new version of the manuscript (L9, L109, L206, Fig. 2).

*Moreover, the way in which 2D horizontal maps are used along the vertical direction should be better analysed. For instance, it would be useful to draw some vertical cross-sections in order to show the effects of the two simulation sets (with and without vertical sampling).*

We agree with the reviewer and proposed to introduce a new figure displaying cross-sections of simulations with and without the sampling approach in order to effectively visualize the effects of the sampling strategy. The new figure 11 is commented in the text lines 437-441.

*In fact the analysis shown in figure 10 is not clear enough.*

We propose to added some information on the dissimilarity index (L428-436) in the revised version of the manuscript.

*Section "3.2 Hard data set" could be improved. (a) There is some confusion between electrofacies and sedimentary facies.*

We understand the confusion between the two terms. We proposed to modify the paper in order to use only the term sedimentary facies, which is more appropriate.

*(b) And what about hydrofacies, which are ultimately the most important for hydraulic conductivity?*

The reviewer raises an interesting remark. The main answer to this question is that two sedimentological units may have similar hydraulic conductivities but very different geometrical shapes. If we associate them to the same hydrofacies during the geostatistical simulation procedure, experience shows that the geometrical shapes of the two sedimentological units gets mixed and loses

consistency. This is why, it is often better to model first the sedimentological units and then fill them with hydraulic properties.

In addition, this process allows to account better for the hard data descriptions and the indirect geological knowledge and sedimental history of the area.

(c) What about lithological logs? Usually they are available if a borehole is drilled for geophysical logs.

The lithological logs are indeed available and used for the sedimentary facies description. We proposed to clarify this point in section 3.2 (L231-242).

(d) Details about the data set, e.g., position and borehole depth, are missing.

The position of the boreholes was indeed missing. This has been corrected and included in figure 1. We also proposed to add some information in the text regarding the borehole data, section 3.2 (L232-236).

Section "3.3 Training images" is not very convincing. It shows that different TIs give different results and some of these are not appropriate with the geological structure of the study area. This is well known and was clearly proved by some of the authors in previous papers. It is well known that the TI should mimic the structures which are expected in the study area and this should be known a priori from geological studies.

The reviewer is correct in stating that the comparison of several TIs is not novel and should be a rather standard step in any MPS simulation study. We do not claim that this is new. But we think that it is important to discuss that aspect in the framework of non stationary TI and simulations. The simulated patterns are often difficult to predict from the TI alone. Previous publications show that complex simulations can be obtained with very simple training images if proper parametrizations and trends are provided. The simulations will be very different from the TI and therefore the selection of the TI requires some trial and error testing.

Therefore, we would like to keep that part which is important for the whole procedure in our opinion. This is a step in which several conceptual models can be tested. A point also mentioned by the reviewer and we agree with him on that. Testing various TI is a step of the workflow. We think that at least to illustrate these ideas, this part of the paper can be useful for some readers and should be kept.

We proposed a slightly modified version of this section in the revised manuscript lines 247-276.

Moreover, further details should be given for figure 2a, which shows a strange sedimentary structure (see specific comment# 14).

As presented in the last comment, all of the three TIs described an alluvial system represented with the same main spatial pattern evolution. The different shapes express within a facies through the three TIs are proposed in order to test and represent at different scales different sedimentological hypotheses (eg: whether the braided river deposit cut or not the alluvial fan facies).

The facies name must not be taken in a narrow sense as they represent more a location of the depositional environment within the whole alluvial system than a facies description.

I am afraid that the term "braided river facies" is probably used in a non rigorous way. In fact, figure 2b shows the typical structure of a "braided river", with a great number of intersecting channels.

We disagree with the reviewer but we did not explain clearly enough the reasoning behind this figure. As explained in the previous answer, it is important to consider different pattern configurations in different training images and check how this will be transferred to the regional simulations when combined with non stationarity parameters and trends. This idea led to the creation of three TIs and three different representations of the braided river facies. In the first TI, the braided river facies represents the entire braided channel belt without describing its internal heterogeneity. In the second and third TI, some internal heterogeneity is included in the concept.

The reason why the entire braided river channel belt can be considered as one single channel is that it happens that there is little potential to preserve low permeability sediments in the braided river channel belt. There is internal heterogeneity in the braided system for sure at a meter scale. But at the reservoir scale, one may consider that the important contrast is the one between the braided river belt and the floodplain. Therefore, it could be reasonable to model the system in that manner.

We proposed a modified version of this in the revised manuscript lines 247-276.

In other words, areas characterised by meandering rivers show a very strong heterogeneity at relatively fine-scale. This is not properly represented by the TIs.

The same explanation stands for the meandering river, where we decided after detailed discussions with the geologists to represent the meandering river belt and not each individual meanders. This approach is explained in the figure 2 e) and is described in detail in the PhD thesis of *Issautier, Benoît. (2011). Impact des hétérogénéités sédimentaires sur le stockage géologique du CO2. University of Aix-Marseille, France : https://www.theses.fr/2011AIX10136.*

Figure 8 shows that high probability for "floodplain" facies determines approximately linear structures. It seems that these structures separate the similar geometrical features observed for "braided river" and "meandering river" facies. Is this right? This seems to be implicitly stated also in the text.

The geological concept and training images imply indeed an alternation of channels (either meandering or braided) and flood plain. When a hard data indicates the presence of one of the facies, it will impose a high probability of occurrence for this facies at the hard data location, but also upstream and downstream since the channel belts have this rather linear structure. The shape will not be exactly linear because tolerance is used for the rotation of the channel belts in the plain and the distance between the channels is not constant in the training image. Once a facies is placed, the geological consistency implies that at some lateral distance the other facies (flood plain or channel belt) will have to be present. Therefore, the general pattern identified by the reviewer is correct but the situation is slightly more complex than simple linear trends.

We updated the text to make the point as clear as possible in the paper lines 398-405.

In individual simulations, "floodplain" facies should be more widely distributed, shouldn't it? Why these maps show a different structure? Is this due to the constraint given by the elongated features for highest probability of "river" facies?

In the individual simulations (for example Fig 7), the flood plain is rather widely distributed. The spacing observed between the river belt in the simulation output (figure 7) corresponds to the indications provided by the geologists on the site. In addition, the overall proportion of flood plain is clearly larger than the channels on every single realization. This is visible in figure 7. In the ensemble of simulations and on the probability maps (figure 8) the flood plain facies has the highest probability of occurrence as compared to the other facies. Therefore, we do not think that the flood plain facies should be more widely distributed.

The orientation is missing in all the figures and the scale length is missing in almost all the figures.

Yes we agree. The figures have been modified in order to add scale length and orientation in the revised version of the manuscript.

**Technical comments :**

Line 28. The acronym "PC" is used for the Continental Pliocene aquifer. Moreover, in a couple of sentences, I was confused and I read PC as "personal computer". I understand that "PC" is probably the correct acronym based on initials of French words, but I think that "CP" would be more appropriate as an acronym for the English name.

We understand the possible confusion for the reader. We decided to not use acronyms in the revised version of the paper when describing the different geological layers and propose to use the term "Pliocene" when referring to the "Continental Pliocene layer". We introduced the terminology at the end of the Geology subsection 2.1 (L144-145).

Line 40. Correct "1974)". / Line 58. Correct "Hu, 2008)". / Line 68. Substitute "," with "."./ Lines 95 to 97. I recommend the authors to carefully follow the international recommendations on the use of SI units and style conventions / Lines 95, 97, 110. Substitute "extend" with "extent" or "extension". / Lines 99, 199, 213, 215, 217, 223-225, 286, 344, 345, 361, 362, 404, 419. I think the use of "meander" as adjective is not correct. I suggest to substitute "meander river" with "meandering river". / Line 104. Substitute "plain itself" with "floodplain". / Line 117. Substitute "of" with "by".

These technicals comments have been corrected in the revised version of the manuscript.

Line 112. Substitute "in" with "at" before "some locations". Rephrase "up to 8m higher on average".

We proposed to change the sentence to : "In the 1960s, the piezometric level was on average 8 m higher as compared to the 2012 data and even artesian at some locations." (L151-155).

Line 166. It is not clear if the TI is a map in the horizontal plane or a vertical cross-section.

As address above in the specific comments section, we explicitly defined that the TIs are horizontal maps view in the new version of the manuscript (L9, L109, L206, Fig. 2).

Line 171. Which is the direction of the x coordinate axis?

The TI is not spatially oriented, however, the x-direction can be assimilated to the east-west direction on the grid.

Lines 196-197. Clarify the expression "By studying the evolution of these response curves".

This has been clarified in the new version lines 236-241.

Line 212. The expression "an analogue river system from northern Italy" does not provide a useful information. Which river? Which kind of geological setting? Moreover, from figure 2a, the braided river facies cover an extended area and does not properly represent the internal heterogeneity of a braided river system.

We agree with the reviewer that the expression "an analogue river system" can be misleading and lack precision. We used satellite images of the Tagliamento river, which is located in Northern Italy near the town of Udine and close to the Slovenia border to create the first TI. We changed the description (L253-257) in the new version of the article

As for the comment on the "braided river facies", this point is already answered above within the specific comment 4.

Line 216. Substitute "meander objects" with "meanders". / Line 319. Substitute "doesn't" with "does not". / Line 340. Substitute "are" with "is".

This is corrected in the revised version of the manuscript.

Lines 285-286. Why "the best way to control the vertical continuity was to sample only from three facies"? Can you comment on this and explain this result?

Since the "floodplain facies" is the most frequent, sampling the facies at random location leads to an over-representation of the flood plain and tends to bias the MPS simulations. After some tests, it appeared that the easiest way to control the connectivity of the objects of interest was to sample only those facies (alluvial fan, braided and meandering river). We also decided to not sample the levee and crevasse splay facies in order to avoid constraining the whole structure of the fluvial objects too heavily.

We added these explanations in the revised version of the manuscript lines 331-336.

We agree and replaced the sentence in the revised manuscript lines 446-447.

Lines 401 to 403. This remark is not so evident from the analysis of the results.

We proposed to clarify the explanations for figure 10 in section 4.4 (L420-428). Moreover, the new figure 11 presents cross-sections through the simulation with and without the sampling approach. We believe that this figure clarifies the effect of the sampling approach on the vertical connectivity of the river beds.

Line 495. Erase "Tectonophysics". /  Figure 3. Substitute "c)" with "b)".

We agree and corrected it in the new version.

**Answer to Referee #2 :**

**Specific comments :**

The starting point for the proposed strategy, as stated in the abstract, is a conceptual model. The conceptual model is considered to be known deterministically. There is no mention of alternative, conceptual models. This is not trivial, and the authors need to justify this approach and suggest how to relax the constraint imposed by using a single conceptual model (see abstract lines 3-5).

We thank the reviewer for pointing out this issue in the abstract. We revised it to mention the possibility of using several alternative training images and conceptual models (L4-5).

Later, in the paper, we already considered three alternative training images (see figure 2), illustrating exactly this point. More conceptual models can be easily included in the workflow and tested.

The strategy presented here is smart in that it assimilates concepts from geology with geostatistical concepts. For example, the use of a physical-mathematical model for establishing the spatial evolution of sedimentary patterns. But some additional work is needed.

We thank the reviewer for his positive comment. Just to be sure to be well understood. The physical model that we solve is used only to get the main trends and obtain plausible patterns. We do not solve the complete coupled system of equations describing the whole processes of sediment transport, deposition, compaction, diagenesis, etc. Our approach is very much simplified as compared to proper sedimentary basin modeling approaches.

I recall some work done along this line (Item 2 above) by Steve Gorelick's group. As I recall, it requires using weather patterns over geological time scales, for boundary conditions. So, different patterns would have a strong effect on the patterns mentioned in (2).

As we indicate above, we do not model all the processes. We just compute a trend map to control the

position of the different types of sedimentary structures in the basin. And then we rely on the multiple-point statistics approach to generate the set of realizations. This is much faster than solving the physical problem and it does not require to provide detailed initial and boundary conditions. Our approach is much simpler but it allows us to generate easily a set of realizations and get some ideas about the uncertainty while the approach based on solving the physics is much more computationally demanding. In addition, the process-based approach is not able to ensure that all the borehole data are honored. Therefore the two methods are very different and complementary.

I am trying to figure out how climate/weather conditions permeated into the modeling of sedimentary patterns. There is a brief and incomplete description of the solution of the diffusivity equation. Just a couple of lines, between Line 170 and Line 175, on the topic. The authors need to shed more light on this aspect of their approach, and to show how the math/physical model used in support of (2) is actually constructed.

As explained above, we do not account for these aspects since we want to model the system in a reasonable manner while remaining as simple as possible.

Line 65: Listing the advantages of the 2010 simulation model, the authors state "...no probability is computed...". Question is: why is that an advantage? What are the positive and negative implications or avoiding probabilistic models?

It is true that we do not explain this aspect in detail in the paper for the sake of brevity.

As explained in detail in the original paper from 2010, the direct sampling technique is a multiple-point statistics (MPS) algorithm that resamples some patterns from the training image in a stochastic manner without computing probabilities. Other MPS algorithms need to estimate the probabilities of the different patterns to produce simulations. Often, this is a problem because to estimate a probability one has to count all possible configurations and check the ones that are compatible with the data. The number of configurations can become extremely large and counting all these different configurations can become a technical limitation in terms of computing time or memory usage. Here, with the direct sampling, we resample some patterns in a manner that we ensure that the probabilities are honored but we do not compute them explicitly. This allows us to consider much more complex situations (more sedimentary facies, for example, a larger size of the patterns, or multivariate patterns) than more traditional MPS algorithms such as SNESIM. All the tests that we have done since 2010 show that this feature is an important advantage as compared to the other MPS methods because it offers much more flexibility. One possible limitation is that the time required to generate a simulation can be larger if the code is not optimized and parallelized. But this is not the case for DeeSse.

To avoid entering into a long discussion, we have added some additional references describing some features of the DeeSse algorithm lines 66-70.

In Sections 4.2 and 4.3 there is a reference to probabilistic models. Confusing, and some clarification is required.

We are sorry that this was confusing. It's true that we do not compute probabilities during the simulation step. But the models are probabilistic. Not computing the probabilities is a technical trick. It does not mean that there is no underlying stochastic process and probabilities. To try to clarify, in

addition to the new references that are provided, we added a sentence explaining that DeeSse is used to generate an ensemble of realizations from which one can estimate any relevant probabilities for the problem of interest (L68-69).

At the top of Section 4.2, the authors state as follows: "Simulating a large number of realizations enables us to calculate probability maps". That is obviously true: when you generate multiple realizations, you can compute probabilities. Question is: what is the connection between these probabilities, on one hand, and uncertainty and risk, on the other? The authors need to make a convincing case that they model uncertainty accurately. Without it, they can only say that they can generate images.

This is a very interesting and important point that has led to heated discussions in the past and that will continue for sure to raise many discussions. The debate goes much beyond the context of this paper. The question of the reviewer revisits the debate about the subjectivist and frequentist interpretations of the notion of probability. This debate has involved mathematicians, philosophers, statisticians, etc. We do not think that it is reasonable to open this debate here since we will not be able to close it for sure.

In short, we consider that we are computing a probability that we interpret as subjectivist. It is a representation of our confidence in the model that we built and the amount of information that we have. We do not claim more than that.

We think that the quality of the uncertainty estimation could be partly tested using cross-validation. This work is not presented in that paper, because our data set is too small and almost all of the models perform equally well (or bad) when there is little data available. If more data would be available, we could certainly compare the local accuracy and the calibration of the predictions of various stochastic models. We plan to do that in the future, but do not have the data for conducting that study yet. This is investigated in the paper currently submitted Juda P. : "*Juda, P., Renard, P., & Straubhaar, J. (2020). A framework for the cross-validation of categorical geostatistical simulations*". We added this reference and this explication lines 481-483.

We still want to add a word of caution, local accuracy and calibration (meaning that we predict correctly the uncertainty on the facies at a certain location) do not necessarily mean that the connectivity of the sedimentary features is well honored and that the groundwater response of the model represents correctly the true one. Therefore, even if we use cross-validation and if performances are good, it may very well happen that the groundwater predictions are not.

To summarize, we agree with the reviewer that the meaning of the estimated probability is an important issue that is not yet fully solved, but we tend to disagree with his last comment since we believe that such methods are useful to bring geological concepts in uncertainty estimations.

We need to see how the innovation (generating sedimentary patterns using a math/physical model) proposed in this study could make a difference. How would the generated images look like without the innovation? How does this innovation help in reducing uncertainty and improving accuracy? Some sort of cross-validation study (comparing results obtained with and without the improvement) could be helpful.

The author raises again an interesting point, however, we do not think that the aim of this paper is to compare the MPS approach against other ones. Other publications have already compared MPS

against SGS or pluri-Gaussian simulations and have shown the benefits of the multiple-point approach and its ability for simulating complex and realistic patterns.

The aim of this work is to present a new workflow that allows to generate complex non-stationary structures at a reservoir scale using MPS when little hard data are available. We propose to modify the introduction to better explain that objective (L102-115).

One of the novel ingredients of the proposed workflow is the computation of the trend using the solution of a diffusivity equation. We proposed to extend the discussion about the advantage of this part of the method in the conclusion (L453-456).

Regarding the advantage of using a trend map created from a math/physical model, we do not think that it would be useful to compare this approach against simulations that do not use a trend map. Indeed, it is well known from previous publications (eg. Chugunova and Hu, 2008) that using a non-stationary training image without accounting for the trend creates some disordered patterns.

Finally, as mentioned above and in the article, the cross-validation is an interesting aspect. But the lack of hard conditioning data makes its application difficult. It is planned in future work to acquire more data and to use cross validation to compare the performance of several geostatistical approaches. For the moment, we still believe that introducing the proposed workflow is interesting because it could be used by other researchers and adapted to their own studies.

**Answer to Referee #3 :**

**General comments :**

This manuscript presents a method for large-scale 3D MPS simulations in areas without the necessity of creating a 3D Training Image, which can be a cumbersome and tedious task. Especially, the presented method focuses on areas with few geological observations and little to no geophysical data (soft conditioning data). Overall, a well written paper, albeit with some technical errors, which can easily be corrected. I recommend the paper for publication. Moderate review.

We are thankful to the reviewer for his/her overall evaluation and its detailed comments on the paper. The specific and technical comments have helped to improve the manuscript.

Below we discuss more precisely the different issues raised by the reviewer and how we adjust the paper in consequence.

**Specific comments :**

Line 21-22: Explain why we want to skip this step. There are researchers who spend a large portion of their time building handheld 3D geological models and would not understand why it is an advantage to bypass the 3D TI.

This is an important point that was not sufficiently well explained in the introduction. We proposed to add a paragraph explaining why we think that this is an important aspect of the methodology lines 87-97.

Line 51-61: A good overview, perhaps also mention Image Quilting Simulation (IQSIM) (Hoffimann 2017 - Stochastic simulation by image quilting of process-based geological models) and other more recent methods if any.

We have added this reference in the revised version of the manuscript, we also added more recent advances made on the DeeSse algorithm (L49-60).

Line 68-69: Slightly elaborate "many options", so that the reader can grasp the advantages/disadvantages of DeeSse better.

We proposed to add recent publication on the code DeeSse but would prefer to no explain/list in detail all of the DeeSse options and parameters (L69-70).

Line 124-125: I think you mean two-point statistics. A variogram is not a geostatistical method, but a mathematical/statistical construct that is used in different geostatistical methods.

We agree on this point and have replaced "variogram" by "two-points statistics" in the new manuscript line 41.

Line 180-181: Making 3D Tis is not "too complex". In fact, in the literature, many studies are presented where handheld 3D geological models are created, which are essentially 3D TIs. Rephrase it to emphasize that it is a difficult, time consuming and subjective way of modeling, but NOT "too complex".

The reviewer is right on the subjective use of "too complex" and that some other studies lead to the creation of 3D geological models, which could be used as TI. We modify the manuscript in consequence line 19.

Line 201: What geological map? You should show it or cite it. If it is the one in Figure 1, then include a reference to figure 1 here.

We added the reference to the cited geological map (L243-245).

Line 204-205: If it is essential to condition the model to the geological map, then you should describe how you do this.

We agree on this point and proposed to add some information on the process lines 243-247 of the revised manuscript.

Figure 3 caption: What is c) referring to? And b) is not mentioned. Needs to be fixed. Also, the grey model makes sense, but you should still describe it in the figure caption.

It was a typo in the caption and modified the caption of the figure 3.

Figure 4: In the b) and c) it would be a lot better if you added a thin black line to mark the outline of the layers, especially in relation to the stacked trend map in c). As for the current state of the figure I

do not really get a lot of information from the stacked trend maps since all the colors of the different layers are identical by design.

The stacked trend map only shows the progradation of the trend as we move upward in the layers, the dark blue color is moving gradually towards the sea as the system evolves. A new 3D view is proposed in the revised version of the figure 4.

Line 256-257: What do the diffusivity equations look like? And what is considered "proper boundary conditions"?

To answer this comment, we rewrite the text in the trend maps section 3.5  (L298-309).

Line 274: Insert a reference to Figure 5b) here.

This has been corrected in the revised version.

Line 276-277: Since river system are highly dynamic in nature, and since the reader has no idea about the sedimentation rates over the last 6 My in the given sedimentary basin, you should probably state why we can safely assume that the variation of the paleo orientations are encompassed by +/- 10 degrees of the values observed at the surface currently.

It is true that river systems are highly dynamic and that their bed orientations can vary through time. However, since our TI encompasses the whole river bed, as shown in figure 2d, we do not expect to see strong orientation changes. The rotation map is derived from interpretation and thus cannot be taken as true fixed values of the paleo orientations. The +/- 10 degrees of tolerance is fixed to take into account this uncertainty. This tolerance also helps to not overconstrain the model and to accommodate the location of the patterns to the hard data during simulations.

We think that the sentence "The orientation map is based on interpretation and therefore uncertain. DeeSse allows to account for this uncertainty. A tolerance of +/- 10° is considered and added/subtracted to the kriged map to obtain two rotation maps..." gives enough information on the reader on the fact that the kriged value is not the true orientation of the paleo river and that some flexibility must be left to the algorithm to simulate the orientation of the patterns.

Line 285-287: Seems a bit discerning, can you elaborate as to why the vertical continuity was not as good when all 6 facies were included for sampling?

This question has already been raised by the reviewer#1. We propose to copy the answer of this comment :

Since the "floodplain facies" is the most frequent, sampling the facies at random location leads to an over-representation of the flood plain and tends to bias the MPS simulations. After some tests, it appeared that the easiest way to control the connectivity of the objects of interest was to sample only those facies (alluvial fan, braided and meandering river). We also decided to not sample the levee and crevasse splay facies in order to avoid constraining the whole structure of the fluvial objects too heavily.

Figure 7: You should always introduce each sub figure in order a) -> b) -> c), not a) -> c) -> b). Also, a very important detail is that nowhere in the paper do you present a vertical slice, or profile, of the simulated models, so the reader cannot see how bad/well the vertical constraints worked in comparison to a full 3D TI. It would be easy to add a cross section view in this figure.

The caption for figure 7 has been corrected, the b) and c) were simply swapped.

We understand the need of cross section view in order to visually inspect the effect of the vertical sampling approach on the simulation outputs. We propose a new figure 11 including these elements. The figure is presented at the end of this document. We also add a small description in section 4.4 (L437-441).

Line 346-348: How do the maps explain that the model is not over constrained? It might be obvious to you, but not necessarily the reader.

We agree with the reviewer that more information needs to be included for the reader. We proposed a revised version for the manuscript (L398-405) providing clearer explanations.

Lines 385-387: I think you are going need to touch on the strengths/weaknesses when making a comparison like that. The so-called classical MPS studies simply have a lot of geophysical/conditioning data available to them, and will, in large, be better conditioned to the actual subsurface. On the other hand, your method is clearly advantageous when you do not have a lot of geophysical/conditioning data available, but of course will not be as nicely conditioned to the actual subsurface. There are many places in the world where they need methods like this, since they do not have elaborate geophysical data sets available to them.

We think that the presented method should not be viewed in opposition to other methods where geophysical data are available. There is no strength in lacking soft information, the objects are not well described neither in their size nor their location and validation approach cannot be performed when lacking conditioning data.

The proposed workflow is only an alternative that tries to take into account most of the available conceptual knowledge.

Line 408-409: How are they satisfactorily reproduced? Based on what? Is it solely based on the boreholes being conditioned correctly and the simulations resembling the TI? Then, it would be nice to show some statistics regarding how well the boreholes and simulations agree.

The facies proportions are satisfactorily reproduced base on the fact that they are similar to the borehole facies proportion distributions, excepted for the alluvial fan proportion that is under-represented in the hard data but compensated with the influence of the TI.

We believe that this is shown in figure 10 a and explain in the sub-section 4.4.

**Specific comments :**

In general, it is not called a "meander river", but a "meandering river". This needs to be fixed.

This has been corrected in the manuscript and in the figures.

Line 28: The abbreviations for Marine Pliocene aquifer is PMS. Perhaps this is an abbreviation that makes sense in relation to the French name for the unit, but in order to make the paper more readable I recommend using MPA, which makes more sense. Line 28: Similarly, the abbreviation for Continental Pliocene is PC. It would be more fitting to use CP.

This comment was already mentioned by the reviewer#1. Here is the proposed answer :

We understand the possible confusion for the reader. We decided to not use acronyms in the revised version of the paper when describing the different geological layers and propose to use the term "Pliocene" when referring to the "Continental Pliocene layer". We introduced the terminology at the end of the Geology subsection 2.1 (L144-145).

Line 68: Change "It" to "it", i.e. no capital letters after comma. / Line 76-77: Add ":" after "The paper is structured as follows", followed by changing "The" to "the". / Line 78: Missing comma after DeeSse algorithm, and "Section 3" should not have a capital letter. / Line 95: You mean the "extent" and not "extend". / Line 238: change "express" to "expressed". / Line 360: Change "the alluvial fan dominate" to "the alluvial fan dominates".

These technicals comments have been addressed in the revised version of the manuscript.

[revised manuscript text omitted]